# Fish-hunting cone snail disrupts prey's glucose homeostasis with weaponized mimetics of somatostatin and insulin

Ho Yan Yeung [1,2], Iris Bea L. Ramiro [1], Daniel B. Andersen[1,3], Thomas Lund Koch[1,2,4], Alexander Hamilton[5,6], Walden E. Bjørn-Yoshimoto [1], Samuel Espino[4], Sergey Y. Vakhrushev[7], Kasper B. Pedersen[7], Noortje de Haan [8], Agnes L. Hipgrave Ederveen [8], Baldomero M. Olivera[4], Jakob G. Knudsen [5], Hans Bräuner-Osborne [9], Katrine T. Schjoldager [7], Jens Juul Holst [1,3] & Helena Safavi-Hemami [1,2,4] ✉

Venomous animals have evolved diverse molecular mechanisms to incapacitate prey and defend against predators. Most venom components disrupt nervous, locomotor, and cardiovascular systems or cause tissue damage. The discovery that certain fish-hunting cone snails use weaponized insulins to induce hypoglycemic shock in prey highlights a unique example of toxins targeting glucose homeostasis. Here, we show that, in addition to insulins, the deadly fish hunter, *Conus geographus*, uses a selective somatostatin receptor 2 (SSTR₂) agonist that blocks the release of the insulin-counteracting hormone glucagon, thereby exacerbating insulin-induced hypoglycemia in prey. The native toxin, Consomatin nG1, exists in several proteoforms with a minimized vertebrate somatostatin-like core motif connected to a heavily glycosylated N-terminal region. We demonstrate that the toxin's N-terminal tail closely mimics a glycosylated somatostatin from fish pancreas and is crucial for activating the fish SSTR₂. Collectively, these findings provide a stunning example of chemical mimicry, highlight the combinatorial nature of venom components, and establish glucose homeostasis as an effective target for prey capture.

Venomous animals have evolved a diversity of toxins to incapacitate prey and defend against predators. Many of these toxins have become valuable tools in basic and biomedical research and have been developed as drug leads, drugs, and diagnostic agents[1]. Within the biodiverse lineages of venomous animals, predatory marine cone snails have provided a vast array of small bioactive peptides, termed conopeptides or conotoxins, that mostly target the prey's nervous and locomotor systems[2]. Each of the ~1000 species of cone snail expresses a unique library of hundreds of peptide toxins, several of which have been shown to act together in so-called "cabals" to concertedly disrupt

[1]Department of Biomedical Sciences, University of Copenhagen, Blegdamsvej 3, DK-2200 Copenhagen N, Denmark. [2]Department of Biochemistry, University of Utah, 15 N Medical Drive, Salt Lake City, UT 84112, USA. [3]Novo Nordisk Foundation Centre for Basic Metabolic Research, Blegdamsvej 3, DK-2200 Copenhagen N, Denmark. [4]School of Biological Sciences, University of Utah, 257 South 1400 East, Salt Lake City, UT 84112, USA. [5]Department of Biology, University of Copenhagen, Ole Maaløes Vej 5, DK-2200 Copenhagen N, Denmark. [6]Department of Clinical Sciences in Malmö, Islet Cell Exocytosis, Lund University, Malmö, Sweden. [7]Copenhagen Center for Glycomics, Department of Cellular and Molecular Medicine, University of Copenhagen, Blegdamsvej 3, DK-2200 Copenhagen N, Denmark. [8]Leiden University Medical Center, Center for Proteomics and Metabolomics, 2333 ZA Leiden, The Netherlands. [9]Department of Drug Design and Pharmacology, University of Copenhagen, Jagtvej 160, DK-2100 Copenhagen, Denmark. ✉e-mail: helena.safavi@utah.edu

molecular targets that are linked within a specific physiological circuit[3–5]. For example, fish-hunting cone snails use a "motor cabal" to disrupt the propagation of action potentials at the neuromuscular junction. Motor cabal toxins include those that block presynaptic voltage-gated calcium channels (Ca$_V$), postsynaptic nicotinic acetylcholine receptors (nAChR), and voltage-gated sodium channels (Na$_V$) on muscle cells[6].

In addition to these neurotoxins, we and others have previously shown that a subset of cone snail toxins mimics hormones to effectively hijack important signaling systems in prey or predators[7–12]. We refer to these toxins as doppelganger peptides, or simply doppelgangers[13]. While the lack of comprehensive genome sequencing has limited the systematic interrogation of the evolutionary origins of doppelganger peptides, several studies on cone snails and sea anemones have revealed that most doppelgangers originate from an endogenous signaling peptide gene that, following recruitment into the venom gland, experiences positive selection and diversification to mimic the related signaling peptide in the target animal[14–16]. For example, we previously showed that cone snails express specialized insulins (con-insulins) to rapidly induce dangerously low blood glucose in prey thereby impairing locomotion and facilitating prey capture[10,17]. Following recruitment from a highly conserved insulin gene expressed in the cone's nerve ring, the venom gene has diversified explosively, experiencing directional selection to target heterospecific insulin receptors in prey and possibly also in predators and competitors[14].

The first species to be shown to use venom insulins was *Conus geographus*, one of only two species known to use a "net hunting" strategy that is characterized by the release of toxins into the water that induce hypoactivity and cause sedation[18] (Fig. 1A). Fish exposed to the released venom appear as under the influence of narcotic drugs which led to the term "nirvana cabal" to describe this group of toxins.

The discovery that the nirvana cabal contained several insulins that comatose fish through inducing low blood glucose[10,19] suggested the existence of other toxins that may act in concert to disrupt normal blood glucose levels (normoglycemia) in prey.

Control of normoglycemia is predominantly accomplished by the opposing actions of insulin and glucagon, two pancreatic peptide hormones that maintain blood glucose within a very narrow range[20]. While insulin induces tissue uptake of glucose to lower blood glucose levels, glucagon stimulates the conversion of glycogen to glucose resulting in an increase of blood glucose. In cases of severe hypoglycemia, glucagon can serve as emergency medication to raise blood glucose[21]. Glucagon is secreted from pancreatic alpha cells, a process that is tightly regulated by the inhibitory peptide hormone somatostatin (SS) which in turn is secreted from pancreatic delta cells. Somatostatin-14 (SS-14) is the major circulatory form of SS that, in human, acts on five different G protein-coupled receptors (GPCRs) of the Gα$_{i/o}$-coupled somatostatin receptor (SSTR) family. Activation of SSTR$_2$ expressed in alpha cells suppresses glucagon secretion whereas activation of SSTR$_5$ expressed in beta cells inhibits the secretion of insulin[22]. Emerging evidence suggests that homologous peptide hormones and receptors also play important roles in maintaining normoglycemia in fish[23,24].

We recently demonstrated the existence of a large family of SS-like toxins (consomatins) in diverse lineages of cone snails[12,16]. Consomatins found in "ambush-and-assess" hunters, fish-hunting cones that inject their venom and then wait in ambush until their prey succumbs to the venomous sting, activate the SSTR$_4$ subtype involved in silencing pain and provide analgesia in several mouse models of pain[12]. Additionally, we noticed the presence of a consomatin-encoding transcript in the venom gland of *C. geographus* that, when synthesized and tested at the five human SSTRs, potently activated the SSTR$_2$[12]. This observation suggested that, in addition to insulins, *C. geographus*

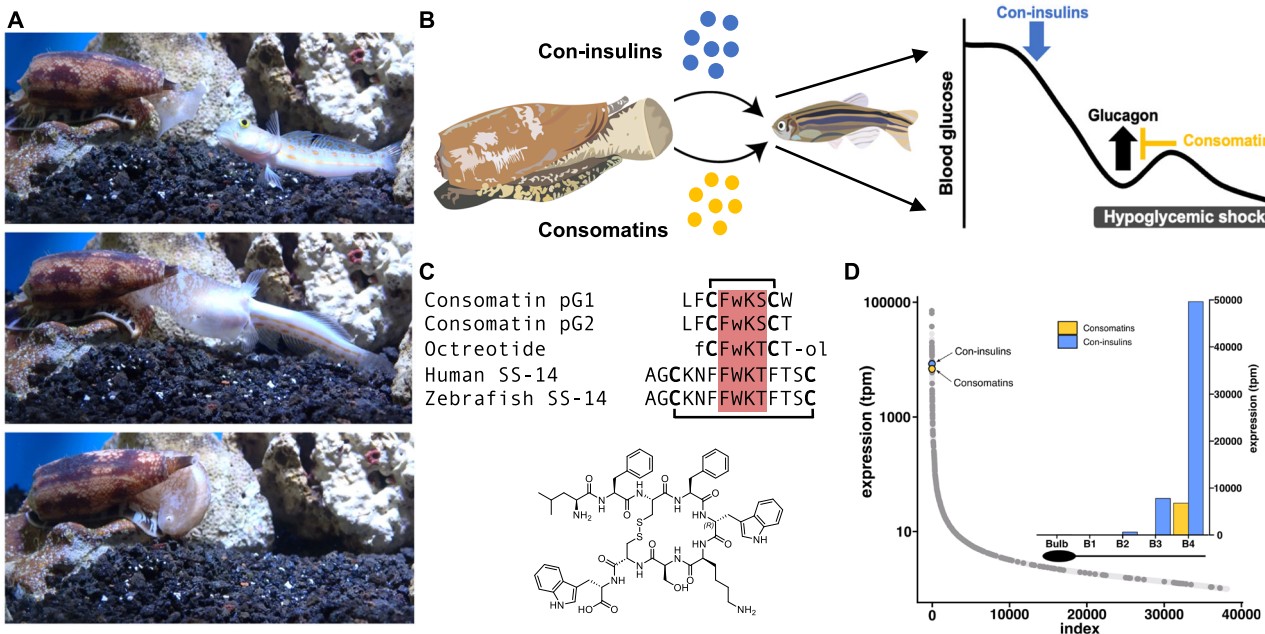

**Fig. 1 | The net-hunting cone snail, *Conus geographus*, uses mimetics of the peptide hormones insulin and somatostatin (SS) to disrupt glucose homeostasis in fish prey. A** Images of *C. geographus* releasing venom into the water to catch fish. Photographs taken by Dylan Taylor. **B** Schematic overview of the central hypothesis: venom insulins (con-insulins) induce dangerously low blood glucose in fish[10]. Consomatins block the secretion of the insulin-counteracting hormone glucagon via potent and selective activation of the somatostatin receptor 2 (SSTR$_2$). Together, these toxins induce sustained hypoglycemia facilitating prey capture. **C** Sequence alignment of the predicted *C. geographus* consomatins, pG1 and pG2,

with the SS drug analog octreotide and human and fish somatostatin-14 (SS-14). Residues important for SSTR activation are highlighted in red. Disulfide bonds are depicted as connecting lines. The chemical structure of Consomatin pG1 is shown below the sequences. Modifications: f, D-phenylalanine; w, D-tryptophan; -ol, alcohol. **D** Con-insulins and consomatins show similar patterns of expression in the *C. geographus* venom gland. Both classes of toxins are some of the most highly expressed transcripts and predominantly expressed in the distal region of the venom gland (segment B4), closest to the pharynx (inset).

may utilize weaponized somatostatins to inhibit glucagon secretion, thereby sustaining dangerously low blood glucose induced by venom insulins (Fig. 1B).

Here, to test this hypothesis, we investigated whether the previously reported *C. geographus* SS-like toxin, Consomatin pG1, could inhibit glucagon secretion in pancreatic isolated mouse islets and perfused rat pancreas. These studies revealed that Consomatin pG1 is indeed able to potently inhibit glucagon secretion in rodents via selective activation of the SSTR$_2$. However, when tested at the homologous fish receptors, Consomatin pG1 only activates one of the two known SSTR$_2$ subtypes suggesting that the chemical identity of the native toxin differed from that of the reported peptide. Indeed, subsequent isolation of the native peptide, Consomatin nG1, from *C. geographus* venom revealed an unusual structure that closely aligns with a SS peptide previously identified from the pancreas of the channel catfish[25]. Investigations into the molecular evolution of the venom peptide strongly suggest that the venom gene experiences directional selection for the fish SSTR$_2$ receptors. The use of weaponized somatostatins and insulins that closely mimic pancreatic hormones expressed in fish prey provides a stunning example of combinatorial chemical mimicry and establishes glucose homeostasis as an effective target for prey capture.

## Results

### Co-expression of con-insulins and consomatins in the *C. geographus* venom gland supports a combinatorial role

Sequencing of the venom gland transcriptome of *C. geographus* previously led to the identification of two transcripts encoding SS-like toxins, referred to as Consomatin G1 and G2[12] (Supp. Fig. S1A). Both transcripts encode toxins that display a cyclic core with residues known to be important for SSTR binding (Phe4-Trp5-Lys6-Ser7, FWKS) and, remarkably, closely resemble the sequence of therapeutic SS analogs such as octreotide (Fig. 1C). This minimized cyclic core is associated with high in vivo stability when compared to the native SS hormone[26]. As previously demonstrated for Consomatin Ro1 isolated from the venom of *Conus rolani*[12], the central Trp5 is likely to be in D configuration. In addition to these two transcripts, we retrieved two variants that only differ by 1-2 amino acids across the entire prepropeptide region (Supp. Fig. S1A).

The venom gland of cone snails consists of a long, convoluted tube that ends in a closed muscular bulb at its proximal site and merges into the proboscis at its distal end. We have previously shown that toxins belonging to the nirvana cabal, including venom insulins, are predominantly expressed in the distal region of the *C. geographus* gland from which they can be released into the water[10]. In this study, to investigate if consomatins and con-insulins are co-expressed in this region of the gland we determined the expression levels in the entire venom gland and in four equal-sized pieces of the gland using RNA sequencing. Closely reflecting the expression pattern of con-insulins, consomatins are most highly expressed in the distal region of the venom gland (labeled as segment B4) suggesting that these toxins may be released together to disrupt glucose homeostasis (Fig. 1D). Similar patterns of expression were observed when interrogating a previously published 454-pyrosequencing dataset[27] (Supp. Fig. S1B). Whether these toxins are expressed from the same glandular cells or whether their release occurs in a coordinated manner remains to be determined.

### Consomatin pG1 is a selective full agonist of the human SSTR$_2$ and shows Gα$_o$ protein bias

To investigate if *C. geographus* consomatins are capable of inhibiting glucagon secretion via selective activation of SSTR$_2$, we synthesized the predicted mature peptide of Consomatin G1, hereafter referred to as Consomatin pG1, for functional evaluation. The mature peptide was predicted based on previous findings of proteolytic cleavage and modification sites of the venom peptide Consomatin Ro1[12].

Consomatin pG1 was synthesized using Fmoc solid phase peptide synthesis and tested at the five human SSTRs using the PRESTO-Tango β-arrestin recruitment assay[28], a transcriptional output-based high throughput assay that has been adapted to quantify the extent of receptor activation upon ligand binding.

Human SS-14 and the SSTR$_2$-selective peptide analog octreotide were used for comparison. SS-14 exhibited full agonism at all hSSTR$_{1-5}$ with nanomolar potencies (Fig. 2A). Octreotide activated SSTR$_2$ more potently than SS-14 (EC$_{50}$ of octreotide and SS-14 are 0.90 nM [pEC$_{50} \pm$ S.E.M. = 9.05 ± 0.04] and 18.47 nM [pEC$_{50} \pm$ S.E.M. = 7.73 ± 0.04], respectively). Similar to other published reports[29], octreotide also activated SSTR$_3$ and SSTR$_5$ although with weaker potencies compared to the activation at the SSTR$_2$ (Supp. Fig. S2). Concurrent with the previously published results on determining the SSTR selectivity of Consomatin pG1[12], here Consomatin pG1 was shown to be a potent and selective SSTR$_2$ full agonist with a potency of EC$_{50}$ = 3.63 nM (pEC$_{50} \pm$ S.E.M. = 8.45 ± 0.06). While the peptide's potency was similar to that of octreotide, Consomatin pG1 is more selective for the SSTR$_2$ than any other SSTR$_2$ agonist reported to date (Fig. 2A, B, Supp. Fig. S2, Table S1).

Having confirmed that Consomatin pG1 selectively activates the hSSTR$_2$, we next interrogated the toxin's G protein signaling profile. The SSTR family is predominantly Gα$_{i/o}$-coupled, leading to inhibition of cAMP production upon receptor activation. The Gα$_{i/o}$ family comprises the Gα$_{i1-3}$, Gα$_{oA}$, Gα$_{oB}$, Gα$_t$ and Gα$_z$ subunits. Gα$_{i1-3}$ subunits are ubiquitously expressed in mammalian tissues, Gα$_{oA}$ and Gα$_{oB}$ are highly expressed in neuronal tissues, Gα$_t$ is primarily expressed in the eyes for sensory functions and Gα$_z$ is found in platelets as well as neurons[30]. Intracellular downstream signaling such as cAMP production is triggered almost instantaneously when the heterotrimeric Gαβγ protein dissociates into Gα and Gβγ subunits upon agonist binding at the GPCR[31]. Gα and Gβγ subunits then interact with their effectors to mediate downstream signaling. To avoid over-amplication of downstream signaling, GPCR kinases (GRKs) phosphorylate the intracellular domain of the active receptor, leading to the recruitment of β-arrestins to facilitate receptor desensitisation. As GRKs have been shown to be one of the effectors of the free Gβγ subunits[32], a panel of bioluminescence resonance energy transfer (BRET)-based split Venus-tagged Gβγ and nanoLuc-luciferase tagged G protein-coupled receptor kinase 2 (GRK2) biosensors[33] were used to indirectly gauge native G$_{i1-3}$, G$_{oA}$ and G$_{oB}$ subunit dissociation from the Gβγ subunits upon receptor activation.

Using this platform, SS-14 mediated potent G protein dissociation at all G$_{i/o}$ subunits tested (Fig. 2C–G, Supp. Fig. S3, Supp. Table S2). Octreotide potently induced dissociation of G$_{i1}$ and G$_{oA}$, which was on par with SS-14 (Fig. 2C, F). pG1 was most potent at dissociating G$_{oA}$ protein among all G$_{i/o}$ proteins tested (Fig. 2F). Notably, when tested at the native human SSTR$_2$ construct, octreotide and pG1 demonstrated weaker potencies at dissociating all G$_{i/o}$ proteins compared to SS-14. The differences in potency between the two different assays may be due to the fact that the Tango-ized SSTR$_2$ construct contains an additional vasopressin receptor 2 tail sequence at its C-terminus which is used to facilitate β-arrestin recruitment[28].

Further G protein bias analysis using the transducer coefficients (ΔLog[τ/K$_A$])[34,35] relative to SS-14 (as the physiological reference ligand) confirmed that, unlike octreotide, pG1 does not bias towards G$_{i1}$ protein but preferentially couples to G$_{oA}$ and G$_{oB}$ protein (Fig. 2H, Supp. Table S3). Together, the results from the PRESTO-Tango β-arrestin recruitment and the G protein dissociation assays suggested that Consomatin pG1 is a SSTR$_2$-selective, G$_o$ subunit-biased peptide that shares a similar structure with octreotide.

### Consomatin pG1 suppresses glucagon secretion in rodents

SSTR$_2$ is highly expressed in both human and rodent pancreatic alpha cells and its activation has been suggested to be critical in suppressing

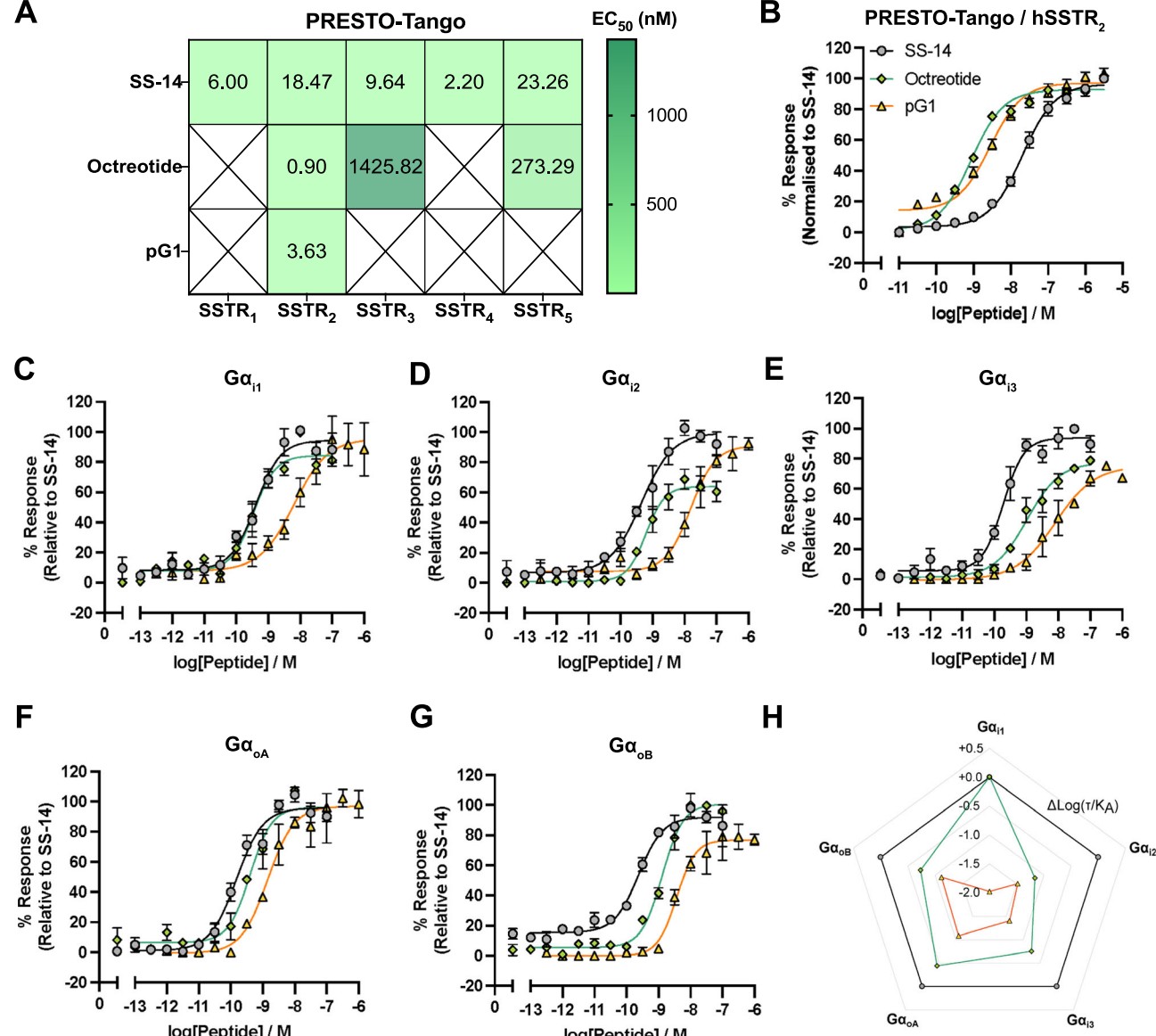

**Fig. 2 | Consomatin pG1 is a potent full agonist of the human SSTR₂ (hSSTR₂) that displays G₀ protein bias. A** Consomatin pG1 selectively activated hSSTR₂ when measured in the PRESTO-Tango β-arrestin recruitment assay. X indicates non-determinable potency values due to a lack of or weak SSTR activation.
**B** Representative dose-response curves of Consomatin pG1 at the hSSTR₂ in comparison to octreotide and SS-14. Error bars represent means ± S.E.M. Experiments were conducted in technical triplicate, $n = 4$ biological replicates. **C**–**G** Ability of

Consomatin pG1 to induce dissociation of $G_{i1}$ ($n = 4$), $G_{i2}$ ($n = 5$), $G_{i3}$ ($n = 4$), $G_{oA}$ ($n = 4$) and $G_{oB}$ ($n = 3$) subunits in comparison to octreotide and SS-14. Representative dose-response curves are shown. Error bars represent means ± S.E.M. All experiments were conducted in technical duplicates. **H** Bias plot in terms of $\Delta Log(\tau/K_A)$ using SS-14 as reference showing that Consomatin pG1 displays G protein bias towards G₀ subunits.

glucagon secretion[22] (Fig. 3A). Given the potent and selective activation of SSTR₂ by Consomatin pG1, we next evaluated if this toxin is capable of suppressing glucagon secretion in the perfused pancreas and isolated islets of Langerhans. Isolated mouse islets were incubated statically with or without 1 nM of Consomatin pG1 or SS-14 for 1 hour in low glucose (1 mM) to induce high basal glucagon secretion, followed by high glucose (10 mM) stimulation. The changes from low to high glucose conditions suppressed basal glucagon secretion by nearly 50% (Fig. 3B). Incubation with either 1 nM SS-14 or 1 nM Consomatin pG1 decreased glucagon secretion by ~ 33% in the 1 mM glucose condition (both $p < 0.01$, 2-way repeated measure ANOVA, Šidák test) (Fig. 3B). Neither SS-14 nor Consomatin pG1 affected glucagon secretion in high glucose conditions.

We next tested the dynamic effect of Consomatin pG1 on glucagon secretion in the perfused rat pancreas over a period of

100 minutes. Consomatin pG1 was administered at three different doses (0.1 nM, 1 nM and 10 nM) for 10 minutes, with 15-minute washout periods in between compound administration. Low glucose (3.5 mM) was administered throughout the perfusion to allow assessment of glucagon suppression induced by Consomatin pG1 at a high basal glucagon level. Similar to the results from the mouse islet studies, 0.1 nM Consomatin pG1 suppressed glucagon secretion by 24% within 5 minute of application (Fig. 3C, D). Glucagon levels failed to return to the average baseline level (i.e., at 754.9 fmol/min) despite the 15-minute washout period, suggesting that Consomatin pG1 may bind strongly or irreversibly to the SSTR₂, potentially requiring significantly longer washout period for Consomatin pG1 to dissociate from the receptor-bound complex. Previous studies demonstrated similar effects of 10 nM of SS-14 on glucagon secretion following a 15-minute washout period[36]. However, as SS-14 also binds to SSTR₅ receptors

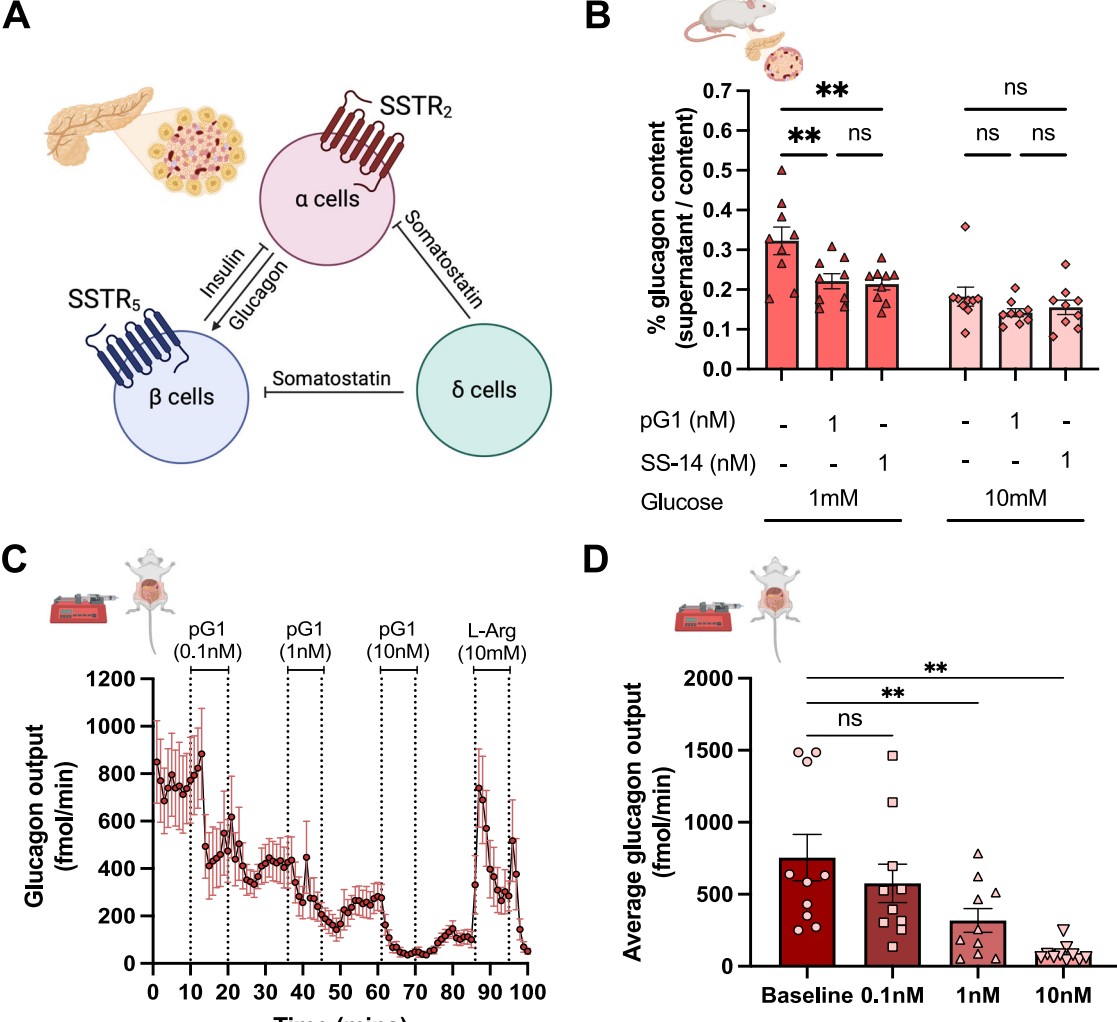

**Fig. 3 | Consomatin pG1 suppresses glucagon secretion in mouse islets of Langerhans and perfused rat pancreas. A** Schematic overview of cell types and SSTR subtypes involved in hormone secretion in pancreatic islets. **B** Consomatin pG1 suppresses glucagon secretion in isolated mouse islets under low glucose conditions (1 mM glucose) ($n = 9$ biological replicates). Data represents means ± S.E.M. ** shows $p < 0.01$ (two-way repeated measure ANOVA, Šidák test). $p = 0.0049$ for comparing control vs pG1 (1 nM) and $p = 0.0024$ for comparing control vs SS-14 (1 nM) at 1 mM glucose condition; n.s.: not statistically significant. **C** 0.1–10 nM of

Consomatin pG1 potently suppresses glucagon secretion in perfused rat pancreas ($n = 10$ biological replicates). Error bars represent means ± S.E.M. **D** Bar plot illustrates the comparison of average glucagon output (fmol / min) of different doses of pG1 with baseline. Data represents means ± S.E.M. measured for $n = 10$ rats. ** shows $p < 0.01$. $p = 0.0032$ for comparing baseline vs pG1 (1 nM) and $p = 0.0061$ for comparing baseline vs pG1 (10 nM) (one-way repeated measure ANOVA, Dunnett's test); n.s.: not statistically significant.

expressed in pancreatic beta cells, a direct comparison between Consomatin pG1 and SS-14 is not possible. Subsequent applications of Consomatin pG1 at 1 nM and 10 nM further suppressed glucagon secretion by 45%. The application of 10 nM Consomatin pG1 drastically reduced the glucagon level close to null (Fig. 3C, D). The rat pancreases were functional throughout the incubation period, as we were able to revert glucagon levels upon stimulation with a bolus dose of 10 mM L-arginine at the end of the experiments (Fig. 3C, D). Consomatin pG1 had no effect on somatostatin or insulin secretion in 3.5 mM glucose (Supp. Fig. S4). Together, our results from isolated mouse islets and perfused rat pancreas demonstrate that Consomatin pG1 is capable of suppressing glucagon secretion likely via selective activation of the SSTR$_2$.

**Consomatin pG1 only activates one of the two homologous fish SSTR$_2$ receptors**
To assess if Consomatin pG1 could activate the homologous receptor(s) expressed in fish we first performed phylogenetic analysis to identify the

corresponding zebrafish (*Danio rerio*) receptor. Consistent with previous studies[37], we found that zebrafish express two isoforms of the SSTR$_2$, sstr2a (Dr-sstr2a) and sstr2b (Dr-sstr2b), that share 77% sequence similarity with each other and 68.0% and 65.1% sequence similarity with the human SSTR$_2$, respectively (Fig. 4A). Related sequences could also be retrieved from various other fish species, including euteleosts that more closely resemble the cone snail's native prey (Fig. 4A). To examine the activity of Consomatin pG1 at the two zebrafish SSTR$_2$ receptors, we designed Dr-sstr2a and Dr-sstr2b constructs for performing PRESTO-Tango β-arrestin recruitment assays. Human SS-14, which is identical in sequence to fish SS-14 (Fig. 1C), was used as a control. While SS-14 potently activated both Dr-sstr2a and Dr-sstr2b (potencies were $EC_{50} = 28.65$ nM [$pEC_{50} ±$ S.E.M. $ = 7.54 ± 0.09$] and $EC_{50} = 131.9$ nM [$pEC_{50} ±$ S.E.M. $ = 6.88 ± 0.17$] respectively), Consomatin pG1 only activated the Dr-sstr2b (potencies were $EC_{50} = 25.32$ nM [$pEC_{50} ±$ S.E.M. $ = 7.60 ± 0.08$] but displayed very low potency at the Dr-sstr2a (Fig. 4B-C, Supp. Table S4). Similarly, pG1 only induced $G_{oA}$ dissociation at the Dr-sstr2b but not the Dr-sstr2a (Supp. Fig. S5, Table S5). However, testing

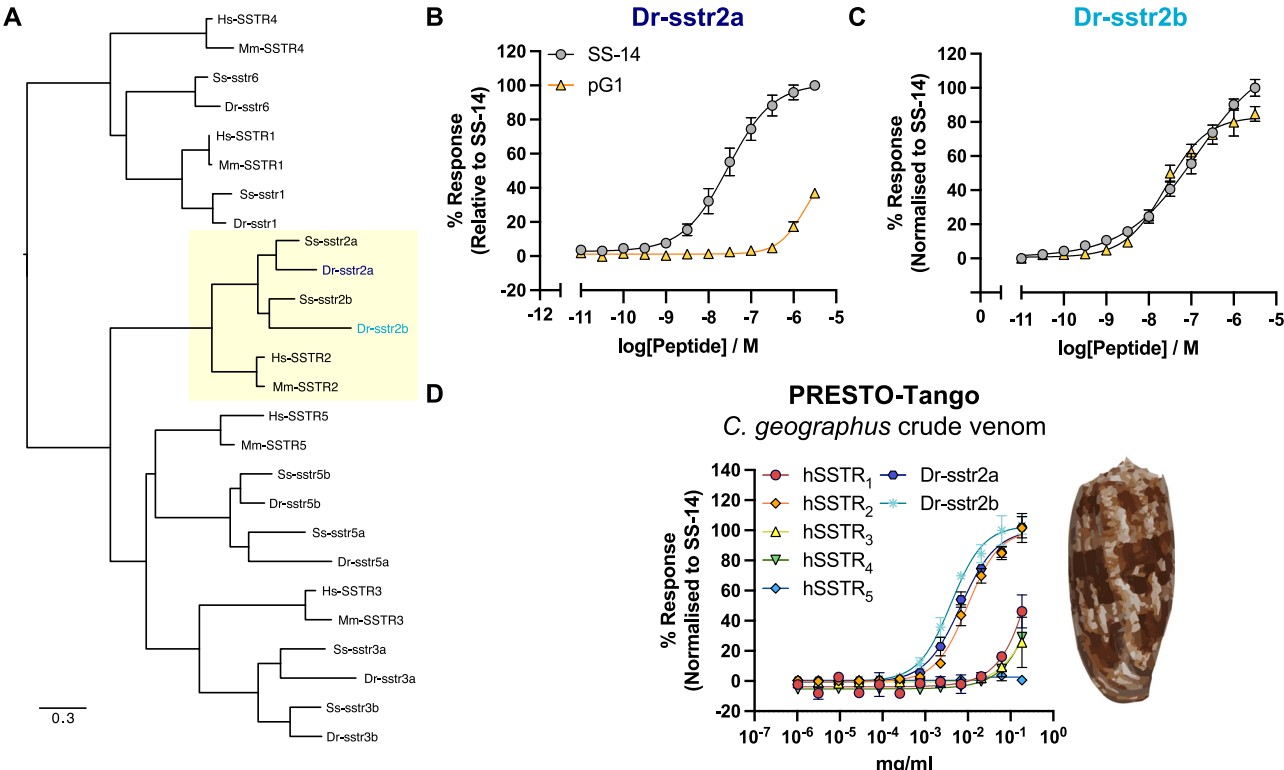

**Fig. 4 | Activity of Consomatin pG1 and crude *C. geographus* venom at the fish and human receptors. A** Gene tree of the human (Hs), mouse (Mm), and fish (*Danio rerio*, Dr; *Salmo salar*, Ss) SSTR subtypes with SSTR₂ highlighted in yellow. **B**, **C** Consomatin pG1 activates the zebrafish Dr-sstr2b ($n = 3$) but not Dr-sstr2a ($n = 6$) isoforms when measured in the PRESTO-Tango β-arrestin recruitment

assay in technical triplicates while **D** crude *C. geographus* venom is capable of activating both Dr-sstr2 isoforms while retaining selectivity for the human SSTR₂. Representative dose-response curves are shown. Error bars represent means ± S.E.M. of $n = 4$ biological replicates. Experiments were conducted in technical duplicates.

venom extracted from the venom gland of *C. geographus* resulted in potent activation of both fish receptors while retaining SSTR₂ selectivity (Fig. 4D, Supp. Table S6). These observations suggested that the native venom peptide contained additional residues or modifications important for Dr-sstr2a activation. Since initial attempts to identify native Consomatin G1 from whole venom using tandem mass spectrometric sequencing (MS/MS) had failed, we used activity-guided assays of fractionated venom combined with MS, MS/MS, and Edman sequencing to identify the native form(s).

### Activity-guided identification of native Consomatin G1 (nG1) from *C. geographus* venom

Extracted *C. geographus* venom was separated into 50 fractions using reversed-phase high performance liquid chromatography (RP-HPLC) (Fig. 5A). Pools consisting of 5 adjacent fractions were then tested for their ability to induce $G_{oA}$ activation at the hSSTR₂. This assay was chosen for screening due to its robustness. Since pool # 6 and # 7 corresponding to fractions 26–35 showed the highest activity (Supp. Fig. S6) we next tested these individual fractions (Supp. Fig. S7). This led to the identification of fraction # 32 as the most potent fraction in inducing $G_{oA}$ activation (Fig. 5B). Further testing using the PRESTO-Tango β-arrestin assay also confirmed the potent activity of # 32 at both hSSTR₂ and Dr-sstr2a (Supp. Fig. S8, Table S7). Fraction # 32 was further fractionated by RP-HPLC, resulting in 30 subfractions that were again tested (Fig. 5C, Supp. Fig. S9). Ultimately, we identified subfraction # 20 (# 32-20) as the most active subfraction. This subfraction potently activated the hSSTR₂, Dr-sstr2a, and Dr-sstr2b while having little activity at other human SSTR subtypes (Fig. 5D). Additionally, we confirmed the ability of this venom fraction to potently inhibit glucagon secretion in perfused rat pancreas (Supp. Fig. S10). Thus, this subfraction was selected for peptide sequencing.

### Consomatin nG1 contains a N-terminal tail that carries diverse *O*-glycans

MS/MS analysis of reduced, alkylated, and trypsin-digested subfraction # 32-20 using an Orbitrap high resolution mass spectrometer identified 3 tryptic peptides corresponding to the Consomatin G1 precursor with a longer N- and C-terminus than the predicted pG1 peptide: Fragment 1 was identified as ZTDVLLDATLLTTOAPEQR where Z is a pyroglutamic acid and O represents hydroxyproline, fragment 2 was identified as LFCFwK where w stands for a potential D-tryptophan as previously identified in Consomatin Ro1, and fragment 3 corresponds to SCWOROYPW-NH₂ (Fig. 6A, Supp. Fig. S11). We further noticed diagnostic HexNAc and HexHexNAc oxonium ions strongly suggesting the presence of glycans in this subfraction (m/z 204.086 and m/z 366.139, respectively) (Supp. Fig. S11A). MS/MS fragmentation using Electron Transfer Dissociation (ETD) enabled the assignment of the glycans to the two threonines in position 12 and 13. Intact mass determination of subfraction # 32-20 identified several MH⁺⁴ precursor ions that corresponded to deconvoluted monoisotopic masses ranging from 5159.404 – 5443.365 (MH⁺¹) (Fig. 6B). The mass shift between several of these ions corresponded to the mass of one deoxyhexose molecule (146.05 Dalton), further suggesting that the native peptide is glycosylated. Reconstruction of the trypsin-digested peptide fragments into a single peptide chain and matching this chain with the tryptic fragments and intact masses observed in this fraction resulted in the following sequence for nG1: ZTDVLLDATLL<u>TT</u>OAPEQRLFC FwKSCWOROYPW, where Thr12 and Thr13 can carry the following five *O*-linked glycan combinations: (Hex)₁ (HexNAc)₁ + (Hex)₁ (HexNAc)₂ (Deoxyhexose)₁, (HexNAc)₂ + (Hex)₁ (HexNAc)₂(Deoxyhexose)₁, (Hex)₁ (HexNAc)₁ + (Hex)₁ (HexNAc)₂ (Deoxyhexose)₂, (Hex)₁ (HexNAc)₁ (Deoxyhexose)₁ + (Hex)₁ (HexNAc)₂ (Deoxyhexose)₁, and (HexNAc)₂ + (Hex)₁ (HexNAc)₂ (Deoxyhexose)₂ (Fig. 6A, B, Supp.

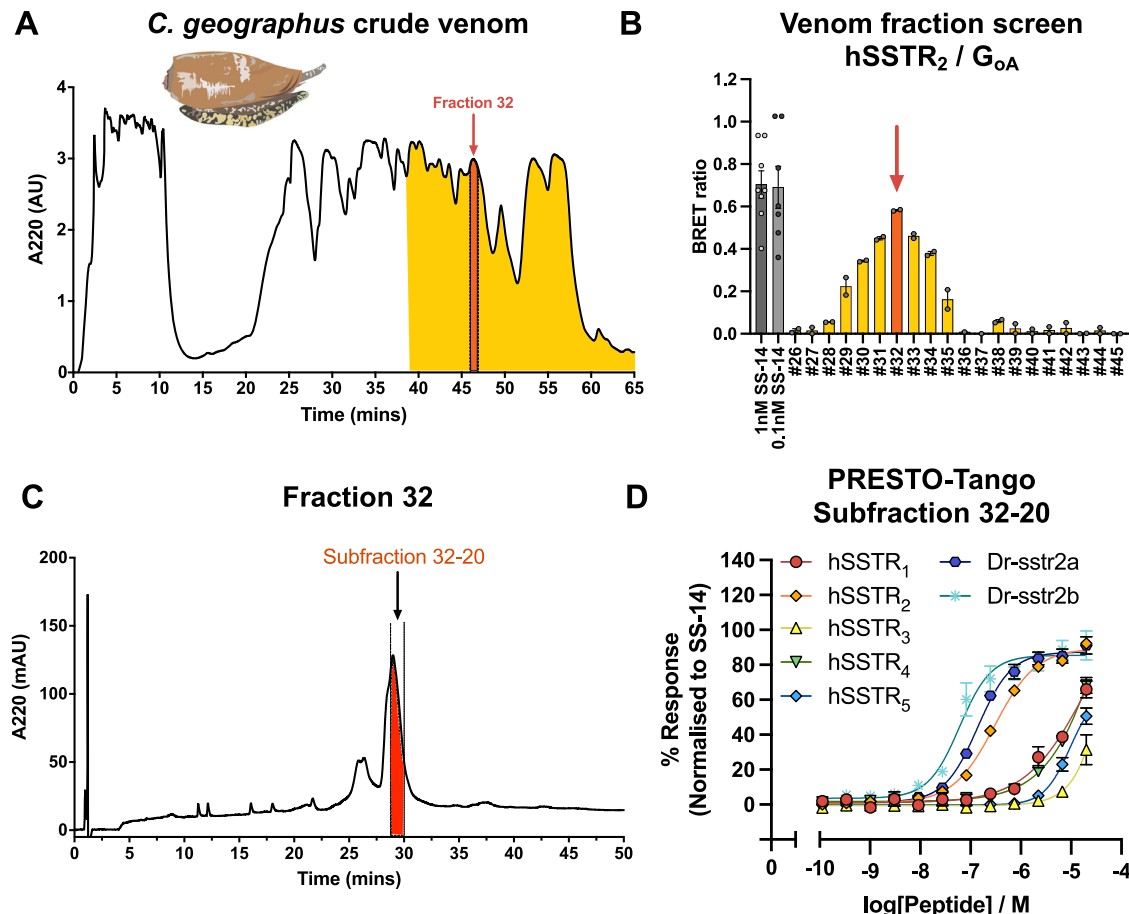

**Fig. 5 | Activity-guided bioassays lead to identification of *C. geographus* subfraction 32-20 as the most potent subfraction in activating hSSTR₂ and fish Dr-sstr2 isoforms. A** Reversed-phase high performance liquid chromatography (RP-HPLC) of *C. geographus* venom used for activity-based identification of native consomatins. **B** Screening of venom fractions identified fraction # 32 as the most potent fraction in activating the $G_{oA}$ subunit at the hSSTR₂. Error bars represent means ± S.E.M. (*n* = 2 biological replicates) in technical duplicates. **C** Further subfractionation using RP-HPLC of fraction # 32 identified **D** subfraction # 32–20 as the most potent in activating hSSTR₂, and both Dr-sstr2a and Dr-sstr2b. Error bars represent means ± S.E.M. (*n* = 3 biological replicates) in technical duplicates.

Fig. S12A). To further confirm the deoxyhexose-containing glycan structures, we performed *O*-glycan release and MS analysis for subfraction # 32-20. In line with the intact mass calculations, and MS/MS analysis of the trypsin-digested samples, several deoxyhexose structures were identified and confirmed by tandem MS (Fig. 6C). Mainly core-1 mucin type *O*-glycans were found to be present in this subfraction. Edman sequencing confirmed the above sequence and supported the presence of modified residues at position Thr12 and Thr13 (Supp. Fig. S13). Furthermore, we were able to identify masses corresponding to different proteoforms of Consomatin nG2 and a tryptic fragment that only differs from Consomatin nG2 by a single amino acid, likely representing an allelic variant of G2 (nG2ii) (Fig. 6B, Supp. Fig. S1A, S12–S14).

Together, these data demonstrate that *C. geographus* consomatins exist in several complex *O*-glycosylated proteoforms. However, future glycomics analysis are needed to identify the exact composition and linkages of the glycans observed for these peptides.

**An analog of Consomatin nG1 lacking *O*-glycans does not exhibit native-like activity**
Protein glycosylations can serve diverse functions, including effects on protein folding, stability, solubility, and biological activity[38]. To investigate the potential role of the complex glycans observed for Consomatin nG1 we tested a synthetic analog of the peptide lacking *O*-glycans (des-glyco nG1). Des-glyco nG1 only partially rescued the potent activation of the Dr-sstr2a observed for the purified venom

fraction, # 32-20 (Supp. Fig. S15, Table S9, potencies were EC₅₀ = 296.4 nM [pEC₅₀ ± S.E.M. = 6.53 ± 0.09] for des-glyco nG1 and EC₅₀ = 1.82 nM [pEC₅₀ ± S.E.M. = 8.74 ± 0.05] for the venom fraction). When tested at the hSSTRs, this analog exhibited slightly lower potencies than the venom fraction while retaining its selectivity for the hSSTR₂ (Supp. Fig. S16). These findings suggest that both the N-terminal extension and glycoslylations are needed for full biological activity. This aligns with previous observations on the role of glycosylations for the biological activity of several other cone snail toxins[8,39]. While glycosylations can also aid in increasing the solubility of peptides in aqueous solutions, as previously demonstrated for the peptide Mg7a from the venom of the giant red bull ant[40], we did not observe any obvious differences in the solubility of des-glyco nG1, pG1, or the venom fraction. However, given the complex nature of the venom peptide we were unable to generate synthetic versions of the glycosylated forms for direct, side-by-side comparison. Thus, whether the observed functional differences between des-glyco nG1 and the purified venom fraction at the Dr-sstr2a are purely based on the presence of the glycans remains to be determined.

**Consomatin nG1 evolved from an endogenous somatostatin-like signaling gene that acquired a novel exon encoding the glycosylated residues**
We previously showed that consomatins evolved from a family of endogenous cone snail neuropeptides called SSRP-like peptides, the protostome orthologs of human somatostatin[16]. Most

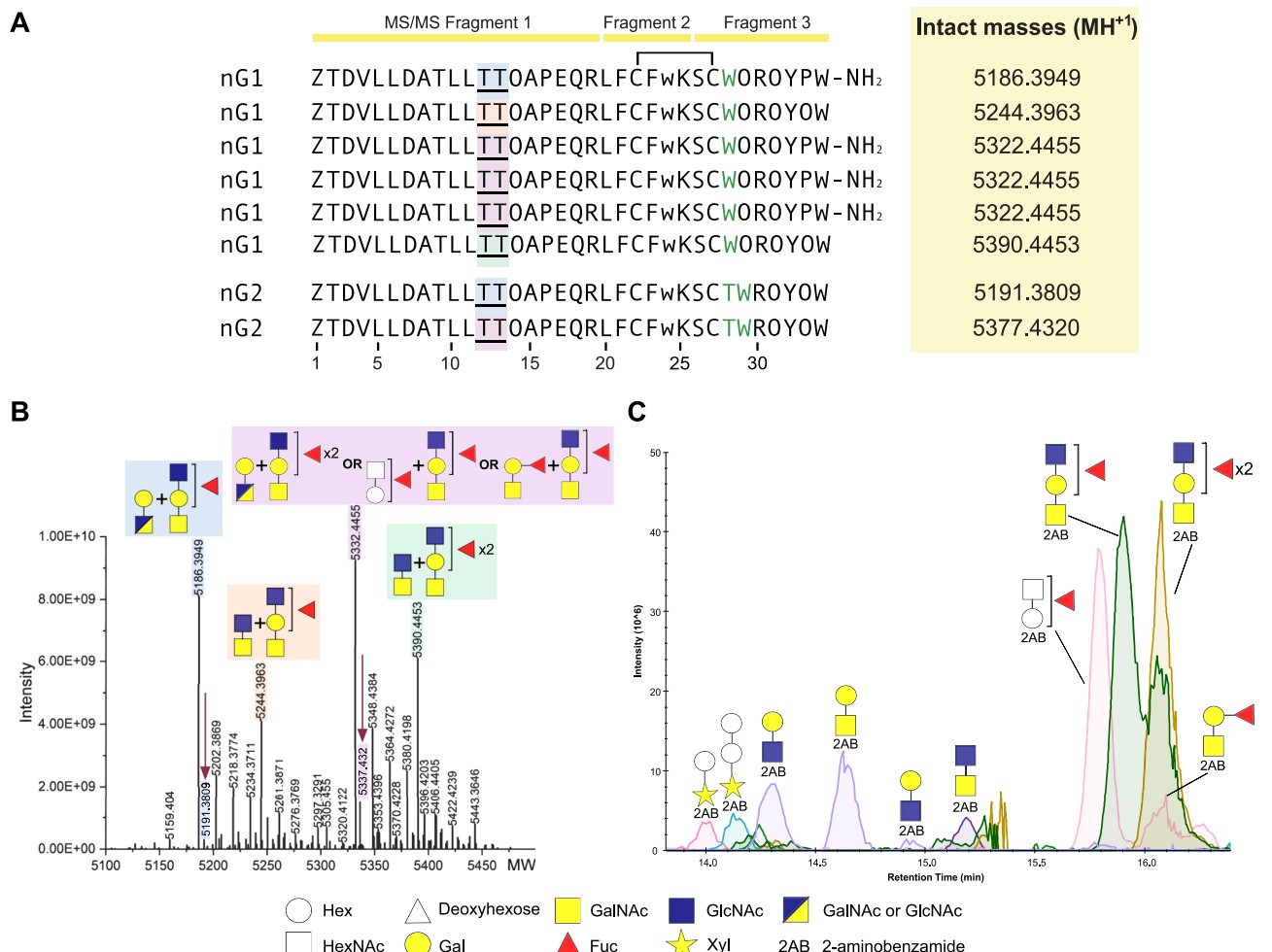

**Fig. 6 | Consomatin nG1 and nG2 exist in several proteoforms that carry various different core-1 glycans. A** Sequences of nG1 and nG2 and intact masses identified in the active venom fraction, # 32-20 by MS, MS/MS and Edman sequencing (see Supp. Fig. S11-S14 for MS/MS spectra, intact mass calculations and Edman sequencing results). Tryptic fragments are indicated above the sequences. Glycan structures correspond to those highlighted in panel B. Residues that differ between nG1 and nG2 are shown in green. Disulfide bonds are depicted as connecting lines. Modifications: Z, pyroglutamic acid; w, predicted D-tryptophan; O, hydroxyproline, T = O-glycosylated threonine. **B** Deconvoluted mass spectrum of precursor ions in

subfraction # 32-20 that correspond to differentially glycosylated proteoforms of Consomatin nG1 and nG2 (indicated by red arrows). Minor peaks correspond to proteoforms containing additional hydroxylations on proline and potentially valine and lysine residues ( + 15.9949, peaks not labeled). Sequences including calculations for modifications are shown in Supp. Fig. S12. **C** LC-MS/MS analysis of released O-glycans after labeling with 2-aminobenzamide (2-AB) revealed that the venom fraction mainly contains core-1 mucin type O-glycans with various degrees of fucosylation, as well xylose (Xyl) and hexose (Hex; possibly mannose) initiated glycan species.

consomatin precursors are of similar length and have highly conserved pro-peptide regions[16]. However, a comparison of Consomatin G1 with other consomatin precursors revealed an unusual extension corresponding roughly to the N-terminal, glycosylated region of the native venom peptides (Fig. 7A). Canonical consomatin and endogenous SSRP-like genes contain 3 exons with the mature peptides encoded on exons 2 and 3[16]. To investigate the origin of the extended region in Consomatin G1 and G2 we interrogated a genomic *C. geographus* fosmid DNA library[14]. Due to the short scaffold length, we were unable to find a single scaffold spanning the entire Consomatin G1 or G2 transcripts. However, scaffold_2839 encoded both the canonical exon 2 of Consomatin G1 and an additional exon located in between the canonical exon 2 and 3. This exon contains a 33-nucleotide stretch encoding the region harboring the glycosylation (ATLL<u>TT</u>PAPEQ) and is separated from exon 2 by a 5,704 bp intron with the canonical splice site motifs (Fig. 7B, Supp. File S1). We therefore conclude that the glycosylated N-terminal extension of Consomatin G1 originated from the de novo birth of a novel exon (exon 3).

### Glycosylated Consomatin nG1 is a motif-minimized mimetic of fish ss4

Given the selective evolution of the N-terminal extension of Consomatin G1 and its importance in receptor activation, we next investigated similarities between the native venom peptides and the endogenous fish hormone(s). Due to several whole genome duplication events, teleost fish express six paralogous somatostatin genes named ss1-ss6[37]. Among the six different ss gene families, peptides derived from the ss4 gene are the most divergent, while peptides derived from the ss1 gene that encodes SS-14 are the most conserved across humans and teleosts. Sequence alignment of Consomatin nG1 with SS sequences retrieved from fish showed similarities within the N-terminal tail region of nG1 to several peptides of the fish ss4 family (Fig. 7C). Interestingly, one of these sequences isolated from the pancreas of the channel catfish, *Ictalurus punctatus*, has been shown to exist in three O-glycoforms[25]. The glycosylated peptides carry a mono-, di-, or trisaccharide on a threonine residue[25] that aligns with Thr12 in Consomatin nG1 (Fig. 7C). Similar sequences with threonine residues in this position could be retrieved from other fish species (Fig. 7C).

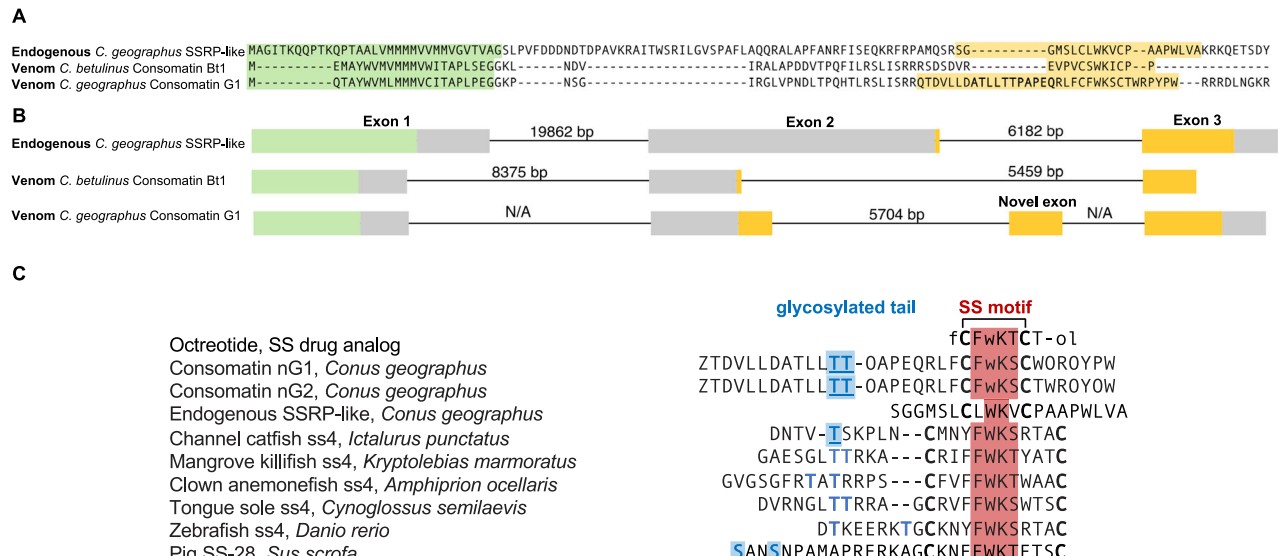

**Fig. 7 | Consomatin G1 evolved from an endogenous somatostatin and related peptides (SSRP)-like gene to mimic both stable SSTR$_2$-selective drug analogs and the glycosylated tail of the fish ss4 peptide family. A** Alignment of the full-length precursors of the endogenous SSRP-like signaling peptide from *Conus geographus*, a canonical consomatin toxin from *Conus betulinus*, and Consomatin G1. Signal sequences are highlighted in green and the mature peptide regions are shown in yellow. **B** Gene structures of the sequences shown in panel B show an additional exon for Consomatin G1 that encodes the extended, glycosylated N-terminal tail. **C** Sequence alignment of the therapeutic SSTR$_2$-selective drug analog octreotide, native Consomatin nG1 and nG2, the endogenous SSRP-like peptide from *C. geographus*, and ss4 sequences retrieved from various fish species, including the glycosylated peptide from the channel catfish. The glycosylated pig SS-28 which has the same sequence as human SS-28 is shown for comparison. Glycosylated residues are underlined and highlighted in blue. Residues that may be glycosylated are depicted in light blue. Amino acids known to be important for the activation of the vertebrate SSTRs are highlighted in red. Modifications, including disulfide bonds, are indicated as shown in Fig. 6.

Whether the corresponding native peptides from other fish species are also glycosylated has not yet been investigated.

Although not aligning with the glycosylation observed in the venom and fish peptide, we note that an *O*-glycosylated form of SS-28, the longer circulating form of SS-14, was recently reported from pig ileum[41] (Fig. 7C), suggesting that this modification may also play a role in activating the mammalian SSTR$_2$ or related receptors. Future structure-activity studies are needed to determine the exact role of the glycosylated tail in SSTR$_2$ receptor activation and signaling.

## Discussion

Tuned by millions of years of evolution, venomous animals express myriads of toxins to incapacitate prey and deter predators. Although doppelganger peptides only represent a subset of this vast chemical diversity, they are proving highly valuable in the design of novel drug leads and biomedical tools and in elucidating the molecular evolution of toxins and endogenous signaling systems. By comparing doppelganger peptides to their endogenous counterparts, one can obtain a unique view into how predator-prey evolution shapes the structure and function of peptides on their trajectory from an endogenous signaling compound to a toxin that targets the homologous system of another organism. This evolutionary transition is typically accompanied by the generation of novel chemical and pharmacological properties that may include improved stability, receptor subtype selectivity, biased signaling, and faster action. Here, we show how this evolutionary path has led to the generation of toxins that contain both, a minimized vertebrate SS-like motif that is nearly identical to the pharmacophore of SS drug analogs, and a N-terminal tail that closely aligns with a fish SS peptide. The similarity to the fish hormone encompasses a complex post-translational modification that is encoded by an additional exon in the *C. geographus* genome. The evolution of Consomatin nG1 and nG2 represents one of the most striking examples of chemical mimicry reported from nature to date. Our observations on the presence of various proteoforms of these peptides suggest potential differences in target selectivity in different species of fish or potential differences in the downstream signaling profiles of these peptides. Future pharmacological studies using synthetic analogs of Consomatin nG1 and nG2 proteoforms are needed to investigate these potential differences.

The discovery of a venom peptide that closely resembles a synthetic SS drug analog exemplifies the potential of natural compounds to serve as alternatives to human-engineered drugs. SSTR$_2$ is the main mediator of SS's antiproliferative effects on normal and cancer cells and has been an intense target for the development of therapeutic, diagnostic, and theranostic agents for various neoplasms and cancers[42,43]. While some neuroendocrine tumors, such as gastrinomas and glucagonomas, predominately express the SSTR$_2$, others co-express two or more SSTR subtypes and are treated using multireceptor-targeted analogs[44]. In light of the structural and functional similarity of Consomatin pG1 to octreotide and other SSTR$_2$-targeting peptides, it stands to reason that, had we identified this doppelganger peptide earlier, we could have bypassed the complex process of designing the first SS drug analogs. These were developed through the application of structure-activity-relationships (SAR) that eventually led to the shortened core binding scaffold seen in octreotide and its subsequently developed single and multi-receptor-targeted analogs[45]. This raises an intriguing question about the extent to which we may uncover similar examples of natural, drug-like compounds in the future, as well as how frequently we unknowingly redesign what nature has already created.

In this context, we previously showed that venom insulins from *C. geographus* and other fish-hunting cones evolved to mimic the native fish hormone while acquiring distinct features tailored to their role as weaponized hormones. Venom insulins lack a segment of the B chain, and in humans and fish, insulin is associated with oligomerization and a slow onset of action[19]. Removing this region in the human hormone

abolishes oligomerization but also results in a near-complete loss of biological activity. Thus, despite decades of research, designing a fully active analog of human insulin that does not oligomerize was never achieved. The discovery of monomeric cone snail insulins capable of activating the human insulin receptor (hIR) enabled the development of several monomeric, fast-acting analogs of human insulin[46,47] demonstrating the value of doppelganger toxins for drug design.

Another example of a doppelganger peptide with drug-like properties that targets glucose homeostasis is exendin-4, a glucagon-like peptide-1 (GLP-1) peptide derived from the salivary gland of the Gila monster, *Heloderma suspectum*[48]. Exendin-4 has a significantly longer in vivo half-life than GLP-1 and, therefore, a much greater potency in lowering plasma glucose[49,50]. Following successful clinical trials, exendin-4 (exenatide) entered the market in 2005 as a first-in-class incretin-mimetic for the treatment of type 2 diabetes (T2D). Since then, several other GLP-1 analogs have been developed and approved for the treatment of diabetes and obesity, such as semaglutide and liraglutide[51]. Interestingly decades after its discovery, the biological role of exendin-4 in the Gila monster remains elusive. While the salivary gland is known to express various toxins, exendin-4 has been suggested to serve endogenous metabolic function rather than acting on the Gila monster's prey or predator[52–54]. While the biological role of exendin-4 remains debated, the presence of insulin and somatostatin-like peptides in *C. geographus* that share little resemblance with the endogenous snail peptides but closely mimic the fish hormones, firmly establishes glucose homeostasis as a target for prey capture.

Although doppelganger peptides have mostly been reported from venomous animals, including cone snails[8–10,12,55], sea anemones[15,56], arthropods[57,58], frogs[59,60], and snakes[61–63], their use is not limited to venoms but has also been reported in parasitic and pathogenic interactions. For example, pathogenic fish viruses utilize viral insulin-like peptides (VILPs) to hijack insulin and insulin-like growth factor (IGF) signaling pathways in their hosts[64,65]. One of these VILPs acts as an antagonist of the human IGF-1 receptor and may provide a new tool to guide the development of analogs for the treatment of IGF-1-sensitive cancers or metabolic disorders[66]. This example highlights the usefulness of doppelganger peptides to reveal novel insight into ligand-receptor interactions. Similarly, venom insulins revealed the existence of a minimized binding motif at the hIR[19,67], a finding that ultimately led to the design of venom-inspired human insulin analogs[46,47]. Here, the existence of *O*-glycosylated threonine residues in *C. geographus* consomatins that closely align with threonines in catfish ss4 and other fish ss4 peptides strongly suggests that this modification plays an important role in activating SSTR$_2$ in fish and potentially in other species.

Our findings have several limitations. Due to the complex structure of Consomatin nG1, identifying the native peptide(s) could only be achieved using activity-based venom fractionations, and MS, MS/MS (HCD and ETD) combined with Edman sequencing. We note that Consomatin nG1 and nG2 had escaped all prior discovery studies on *C. geographus* venom despite decades of research using various orthogonal techniques, including state-of-the-art MS/MS sequencing[8,68–71]. Future studies are needed to determine the exact composition and linkages of the glycans identified here.

We further point out that we do not provide direct evidence for the combinatorial action of con-insulins and consomatins in fish or mouse. Vertebrate animals that experience insulin-induced hypoglycemia need to be rescued with glucose[17] making it difficult to assess consomatin-induced continuation of hypoglycemia in vivo. However, the well-established role of the insulin-glucagon axis in maintaining normoglycemia combined with the data presented here makes it highly likely that these two toxins concertedly act to disrupt glucose homeostasis.

In addition to regulating pancreatic glucagon secretion, SS-mediated activation of SSTR$_2$ is known to suppress the secretion of growth hormone (GH) from the pituitary[72]. Circulating GH increases glucose production through gluconeogenesis and glycogenolysis from the liver and kidney[73]. This effect is partly due to increased levels of insulin-like growth factor 1 (IGF-1) following GH secretion. Interestingly, while SS-14 is a potent inhibitor of GH secretion in fish, the catfish ss4 peptide does not appear to have any GH secretotropic effects[74]. Whether this is due to structural and functional differences related to the glycosylated N-terminal tail has not been addressed but warrants future investigation.

Finally, our findings on the combined use of two classes of weaponized hormones suggest that the nirvana cabal may include additional toxins that target normoglycemia. Such toxins could include incretin-like peptides that act via modulation of the GLP-1, glucose-dependent insulinotropic polypeptide (GIP) or glucagon receptors, toxins that depolarize pancreatic beta cells to induce insulin secretion or those that regulate blood glucose via yet unknown molecular mechanisms. Future identification of these potential peptides may provide novel drug leads and enhance our understanding of the complex physiological mechanisms guiding glucose control.

## Methods
Animal experiments were approved by the Danish Animal Experiments Inspectorate (License Number 2018-15-0201-01397) according to guidelines set by the Danish legislation and the European Union Directive.

### Chemical reagents
Unless otherwise specified, all chemical and cell culture reagents were sourced from Sigma and ThermoFisher. SS-14 and octreotide were obtained from GenScript. Consomatin pG1 and des-glyco nG1 were custom-synthesized by GenScript. The mass and purity ( > 95%) of the peptides were verified by RP-HPLC and MS prior to use.

### Transcriptomic analysis
Transcriptome sequencing was performed on 4-equal size sections of the *C. geographus* venom gland. Sequencing, assembly, annotation and determination of expression levels were obtained as previously described[12]. Sequencing data has been deposited in the NCBI Short Sequence Archive (SRA) under the following accession number: SRR27535905 Section B4, SRR27535906 Section B3, SRR27535907 Section B2, SRR27535908 Section B1.

### Expression analysis
Total RNA was extracted from four segments of the venom duct (B1-B4) of one specimen of *C. geographus* collected in Cebu, the Philippines using the Direct-zol RNA extraction kit (Zymo Research) according to the manufacturer's instructions. Collection was done under the gratuitous permit (0111-16) issued by the Bureau of Fisheries and Aquatic Resources to the University of the Philippines Marine Science Institute. Library preparation and sequencing was performed by the University of Utah High Throughput Genomic Core Facility on Illumina HiSeq2500 as previously described[17]. Raw reads were trimmed of adaptors using fqtrim (v0.9.4) and quality was assessed and filtered using prinseq-lite. Error-correction was performed using Bbnorm ecc tool, and the trimmed and error-corrected reads were assembled using Trinity (v2.2.1) with 31 k-mer length and minimal coverage of 10. Transcript expression was calculated using RSEM. We compared the expression of insulins and consomatins in the four venom gland segments and a previously published RNAseq dataset from a *C. geographus* venom bulb.

We further investigated the expression of consomatins and insulin genes in venom duct segments from previously published RNAseq from 454 GS FLX Titanium sequencing (SRX151239-SRX151242). We mapped the reads to the transcripts using bowtie2 with local alignment and score-min set to G,60,6. The resulting sam-file was sorted

and bam index statistics calculated using samtools. Reads per kilobase per million mapped reads (RPKM) were calculated from the statistics and visualized for each segment.

### DNA constructs

3xHA-tagged human $SSTR_2$ construct was obtained from cDNA.org (cDNA resource center, Bloomsburg PA, US). Constructs, namely masGRK3ct-nLuc, Venus 156-239-$G\beta_2$, Venus 1-155-$G\gamma_1$ and native $G_{i/o}$ proteins, used for the G protein dissociation studies were kind gifts from Prof. Kirill Martemyanov (University of Florida, US). Tango constructs for human $SSTR_{1-5}$ were kind gifts from Prof. Bryan Roth (University of North Carolina, Chapel Hill, US) (Addgene kit # 1000000068). Tango constructs for fish Dr-sstr2a (sequence accession number: XP_005170178.1) and Dr-sstr2b (sequence accession number: XP_017210358.1) were custom-synthesized by Twist Biosciences using the Tango construct template as described in Kroeze et al., 2015[28]. 3xHA-tagged Dr-sstr2a and 3xHA tagged Dr-sstr2b constructs were inserted into pcDNA3.1 backbone. All DNA constructs were sequenced and confirmed prior to use.

### Mammalian cell culture and transfection

HTLA cells were given by Prof. Bryan Roth (University of North Carolina, Chapel Hill, US) and were maintained in Dulbecco's modified Eagle's medium (DMEM) supplemented with 10% Fetal Bovine Serum (FBS) (Biowest) and 100 U/mL penicillin and 100 µg/mL streptomycin at 37 °C in a humidified incubator supplemented with 5% $CO_2$. Constant selective pressure for tTA-dependent luciferase reporter and a β-arrestin 2 tobacco etch virus protease fusion gene were maintained with the addition of 100 µg/mL hygromycin B and 2 µg/ml puromycin (Tango growth medium).

HEK293T cells were given by Prof. Hans Bräuner-Osborne (University of Copenhagen, Denmark) and were cultured in DMEM supplemented with 10% FBS (Biowest) and 100 U/mL penicillin and 100 µg/ml streptomycin at 37 °C in a humidified incubator supplemented with 5% $CO_2$.

### PRESTO-Tango β-arrestin recruitment assay

The assay was done as described in ref. 12. Briefly, HTLA cells (1 million cells/well) were seeded in 6-well plates in Tango growth medium, and incubated overnight. On day 2, medium was changed to Tango growth medium without hygromycin B and puromycin. Cells were then transfected using Polyfect (Qiagen) according to manufacturer's protocol. On day 3, 15,000 cells in 40 µL DMEM supplemented with 1% dialyzed FBS (Tango assay medium) per well were seeded in poly-D-lysine-coated white clear-bottom 384 well plates (Corning) and incubated overnight. On day 4, the Tango assay medium was changed and 10 µL of each of the peptide was added at 5x final concentration. The peptides were resuspended in HBSS with 20 mM HEPES, 1 mM of $CaCl_2$, 1 mM of $MgCl_2$, pH adjusted to 7.4 with 1 M NaOH (Tango assay buffer), which was supplemented with 0.1% BSA (Tango ligand buffer). After overnight incubation, on day 5, the medium and the compounds were removed from the cells and 20 µL of 1:20 dilution of BrightGlo (Promega) in Tango assay buffer supplemented with 0.01% pluronic F68 was added to the cells. Plates were incubated for 20 minutes in the dark at room temperature. Luminescence was measured on a Molecular Devices SpectraMax iD5 (Molecular Devices) with each well integrated for 1 second.

### G protein activation assay

The following protocol was adopted from Masuho et al., 2015[33]. HEK293T cells were seeded onto 6-well plates at a density of 750,000 cells/well. Cells were transfected the next day with 3xHA-tagged $hSSTR_2$ (0.33 µg), $G_{i/o}$ protein (0.66 µg), Venus 156-239-$G\beta_2$ (0.33 µg), Venus 1-155-$G\gamma_1$ (0.33 µg) and masGRK3ct-Nluc (0.33 µg) (at a ratio of 1:2:1:1:1 and the total DNA amount per well was 2 µg) using linear polyethyleneimine (PEI) at 25,000 molecular weight (Polysciences. Inc) with DNA: PEI at 1:6 ratio. 16 to 24 hours post-transfection, HEK293T cells were seeded onto white 96 well plate (PerkinElmer) coated with poly-L-lysine (PLL) at a density of 40,000 cells/well and were allowed to grow overnight. On the day of assay, cells were washed once with phosphate-based saline (PBS) and 80 µL of assay buffer which composed of PBS with 0.5 mM $MgCl_2$ and 0.1% glucose, pH 7.4, were added onto each well. 10 µL of Nluc luciferase substrate furimazine (Promega) were added onto each well and incubated for 3 minutes. 10 µL of 10 x ligands were then added manually onto the plate and BRET signal was measured instantaneously using Molecular Device iD5 plate reader equipped with 485 (20 nm bandwidth) and 535 (25 nm bandwidth) filters with integration time of 500 ms continuously for 15 min.

### Animal husbandry

Male Wistar rats ( ~ 250 g) were housed in a 12:12 hour-light:dark cycle and acclimatized for a week before experiments. C57BL/6 J mice were purchased from Janvier labs (Janiveier, France) and all mice used were between 12-20 weeks of age. All mice were group housed on a 12:12 hour-light:dark cycle with ad libitum access to standard chow and water.

### Glucagon secretion from isolated islets

Female C57BL/6 J mice of 12–20 weeks were used for the islet secretion. Mice were killed by cervical dislocation and pancreatic islets were isolated as previously described[75]. Whole islets were then manually picked and incubated in RPMI 1640 Glutamax media supplemented with 11 mM glucose, 10% FBS and 100 U/ml penicillin and 100 µg/mL streptomycin at 37 °C in a humidified incubator supplemented with 5% $CO_2$ for an hour. For each condition tested, 10 islets were grouped and were tested in triplicates first in 1 mM glucose Krebs-Ringer bicarbonate buffer (KRB) (25 mM $NaHCO_3$, 115 mM NaCl, 3.5 mM KCl, 0.5 mM $MgSO_4$, 0.5 mM $NaH_2PO_4$, 5 mM HEPES, 2.5 mM $CaCl_2$ supplemented with 0.1% BSA free from non-essential fatty acids) for one hour. The islets were subsequently incubated with or without drugs in 10 mM glucose KRB for an hour. Supernatant from both incubations were collected. Total islet glucagon content was determined by acid-ethanol extraction. Both the supernatant and the acid-ethanol content were stored at −80 °C until further analysis. Glucagon in both supernatant and content were measured using U-plex metabolic 2-plex combo (mouse insulin and glucagon) kit (Meso Scale Diagnostic; catalog no: K15303K) according to manufacturer's instruction.

### Rat pancreas perfusion

Non-fasted rats were anesthetized by a subcutaneous injection of a balanced mix of hypnorm and midazolam (0.3 mL/100 g body weight). Rats were placed on a heated operating table (37 °C) and the abdomen were opened. Blood vessels were tied off before removing the large and small intestines, the stomach, and the spleen. Then, the kidneys were ligated. A catheter was inserted into the abdominal aorta, ensuring perfusion of the pancreas through both the coeliac trunk and the superior mesenteric arteries. The perfusion buffer was a modified Krebs-Ringer bicarbonate buffer with 5% (w/v) dextran T-70 (Pharmacosmos, Holbaek, Denmark), 0.1% (w/v) BSA (fraction V), 3.5 mM glucose and 3 mM vamin (Fresenius Kabi, Copenhagen, Denmark), pH adjusted to 7.4. The perfusion buffer was gassed with 95% $O_2$ and 5% $CO_2$ and perfused at a slightly higher than physiological flow rate of 5 mL/min at 37 °C, to ensure the organ had an adequate oxygen supply. Once perfusion of the pancreas was established, a catheter was inserted in the vena portae to collect the venous effluent and the rat was sacrificed by perforation of the diaphragm. After a 30-minute equilibration period, samples were collected every minute for the duration of the protocol. Consomatin pG1 and an aliquot of the lyophilized

venom fraction # 32-20 were resuspended in perfusion buffer with 5% DMSO and co-administered at 0.25 mL/min with the perfusion buffer for 10 minutes at increasing concentrations (0.1 nM, 1 nM, and 10 nM for Consomatin pG1 and 0.1 nM and 1 nM for # 32–20) with a 15-minute (for pG1) and 25-minute (for # 32–20) perfusion buffer wash after each concentration. 10 mM L-Arginine was used as a positive control. Collected samples were immediately transferred on ice and stored at −20 °C until hormone analysis. Glucagon, insulin, and somatostatin concentrations in venous effluents were quantified by use of validated in-house radioimmunoassays (RIA) as previously described[36] (see below). Statistical analysis were performed where appropriate using one-way analysis of variance (ANOVA) with Dunnett's post-test. Statistical significance was taken as $p < 0.05$.

### Biochemical measurements of perfusion effluents

Glucagon concentration in perfusion samples were measured by a validated in-house RIAs using the 4305 antibody, which reacts with the C-terminal epitope of glucagon[36] and no other cleavage products of pro-glucagon. Insulin concentrations were measured with the 2006 antibody, which measures all isoforms of rat insulin. Somatostatin concentrations were measured using the 1758 antibody, which detects both SS-14 and SS-28. Standard curves were generated using synthetic peptides: glucagon 1-29, somatostatin-14, (Bachem, Bubendorf, Switzerland) and insulin (Actrapid; pro.medicine.dk). Monoiodinated $^{125}$-I-labeled glucagon, somatostatin, and insulin were used as tracers. Assay sensitivity was <1 pmol/L ( < 5 pmol/L for insulin detection), allowing reliable measurements of increments of at least 25 fmol/min.

### Statistical analysis

Data analysis were performed in GraphPad Prism (v.9.4.1). Statistical analysis were performed using two-way repeated measure analysis of variance (ANOVA) with Šidák test for isolated islets glucagon secretion and one-way repeated measure ANOVA with Dunnett's test for rat pancreas perfusion experiments. Statistical significance was taken as p < 0.05.

### Receptor phylogeny

Amino acid sequences of somatostatin receptors from *Homo sapiens, Mus musculus, Danio rerio* and *Salmo salar* were downloaded from NCBI Genbank. The sequences were aligned using mafft (v7.520) using local pairwise alignment at maximally 1000 iterations. We constructed a maximum likelihood tree using IQ-tree (v2.2.2.3) (seed: 680031) with the substitution model set to Q.mammal+I + G4 based on Bayesian Information Criterion. Bootstrap values were estimated using IQtree's UF boot with 1000 replicates.

### Molecular evolution of G1

We identified genomic segments encoding the Consomatin G1 gene and gene structure using exonerate (v2.76.2) on a draft genome of *C. geographus*[14]. By querying the *C. geographus* genome with the G1 transcript using the est2genome model, we identified a scaffold encoding complete exon 2 and a complete, additional exon 3 only observed in *C. geographus* Consomatin G1.

### BRET data analysis

BRET ratio was determined by dividing the BRET signal at 535 nm by that at 485 nm. The net BRET ratio was determined by subtracting the BRET ratio of drugs with the BRET ratio of vehicle control. Net BRET ratio over time was then plotted using GraphPad Prism (v.9.4.1), and the area-under-curve of net BRET ratio over time course for each ligand concentration tested were then determined and plotted against log(-concentration) of drugs tested. Results of test ligands were normalized as percentage of maximal SS-14 response. Concentration response curves were fitted with a 4-parameter logistic Eq. (1) using GraphPad

Prism (v.9.4.1):

$$Y = \frac{E_{min} + (E_{max} - E_{min})}{1 + 10^{(\log EC_{50} - X) \times \text{Hill slope}}} \quad (1)$$

### Quantification of ligand bias

Ligand bias were quantified using the method described in[35]. SS-14 was chosen to be the physiological reference ligand given its physiological significance as well as its full agonism at $SSTR_2$. The operational model (2)[35] was used to determine the transduction ratios ($\tau/K_A$).

$$E = E_{min} + \frac{(E_{max} - E_{min})}{1 + \left(\frac{(1 + \frac{[A]}{10^{\log K_A}})}{10^{\log R}[A]}\right)^n} \quad (2)$$

where E is the effect of the ligand, [A] is the concentration of test ligands, $E_{max}$ is the maximal response of the system, $E_{min}$ is the minimum response of the system, $\log K_A$ is the logarithm of the equilibrium dissociation constant, n is the slope of the transducer function, log R is the logarithm of the transduction ratio ($\tau/K_A$).

The operational model was manually input in GraphPad Prism (v.9.4.1) according to van der Westhuizen et al., 2014[35]. Each experiment was analyzed using the operational model which all individual experiments were fitted with a global shared slope. The LogR (equivalent to log[$\tau$]/$K_A$) values were estimated by GraphPad Prism (v.9.4.1) and were used to calculate the relative efficacy (RE) using Eqs. (3) and (4).

$$\Delta \log\left(\frac{\tau}{K_A}\right) = \Delta \log\left(\frac{\tau}{K_A}\right)_{SS-14} - \Delta \log\left(\frac{\tau}{K_A}\right)_{ligand} \quad (3)$$

$$RE = 10^{\Delta \log\left(\frac{\tau}{K_A}\right)} \quad (4)$$

S.E.M. of $\Delta \log\left(\frac{\tau}{K_A}\right)$ were calculated using the following Eq. (5):

$$S.E.M._{ligand} = \sqrt{S.E.M._{SS-14} + S.E.M._{test\ ligand}} \quad (5)$$

### Venom extraction and fractionation

Specimens of *C. geographus* were collected from the Central Philippines. Crude venom (300 mg) was extracted in 20 mL of 40% acetonitrile (ACN) / 0.1% trifluoroacetic acid (TFA) in water. The suspension was vortexed for 1 min and homogenized using a 50 mL glass-teflon homogenizer. The homogenate was vortexed and centrifuged at 15,000 × *g* for 15 min at 4 °C. The supernatant was separated from the pellet and the pellet was extracted again in 20 mL of 40% ACN / 0.1% TFA in water. The supernatants were pooled and diluted with 0.1% TFA to give a final concentration of ~ 10% ACN / 0.1% TFA. The diluted extract was loaded onto a Vydac preparative reversed-phase $C_{18}$ HPLC column (22 × 250 mm, 10–15 μm, 300 Å). The venom components were eluted using 0.1% TFA (buffer A) and 90% ACN / 0.1% TFA in water (buffer $B_{90}$) with a gradient of 1.3% $B_{90}$/min at a flow rate of 20 mL/min. Fractionation was monitored at 220 nm and 280 nm. Individual fractions were collected and stored at -20 °C until further use. Fractions were further subfractionated using a Vydac $C_{18}$ monomeric column (4.6 × 250 mm, 5 μm, 300 Å) and the components eluted using a linear gradient of 25% - 50 % $B_{90}$ for 25 minutes at a flow rate of 1 mL/min. The homogeneity of the active component was verified by another round of HPLC using a gradient of 15% - 62% $B_{90}$ for 47 min at a flow rate of 1 mL/min. The same subfractionation procedure was repeated using additional *C. geographus* venom to isolate additional material that was equivalent to # 32-20 (data shown in Figs. S10, S15, and 16).

## Determination of peptide concentration of fraction # 32 and subfraction # 32-20

Absorbance at $\lambda = 280$ nm of fraction # 32 and subfraction # 32–20 were measured with UV-vis spectrophotometer using a 10 mm pathlength quartz cuvette. The concentration of peptide in the sample was calculated using the following Eq. (6):

$$C = \frac{A}{L(n_Y + n_W)} \qquad (6)$$

Where C is the concentration of the peptide, A is the absorbance measured at $\lambda_{280}$ nm, L is the pathlength (10 mm), n is the number of tyrosine (Y) or tryptophan (W) present in the nG1 sequence and $\in$ is the extinction coefficient of tyrosine or tryptophan.

## Tandem mass spectrometric sequencing

An aliquot of the lyophilized venom fraction containing approximately 2 nmol was reduced with 5 mM of Tris(2-carboxyethyl)phosphine hydrochloride (TCEP) for 15 min at 55 °C and alkylated with 20 mM iodoacetamide (IAA) for 30 min at room temperature. The reduced and alkylated sample was digested by adding trypsin at an approximate enzyme/protein ratio of 1:100. Peptides were cleaned using StageTips prior to MS/MS analysis on a Tribrid Eclipse mass spectrometer (Thermo Fisher Scientific). MS/MS spectra were analyzed using Byonic software (v2.16 Protein Metrics) and manually verified.

## Intact MS analysis

The analysis of subfraction # 32-20 was performed by EASY-nLC 1000 UHPLC (Thermo Scientific Scientific) interfaced with PicoView nanoSpray ion source (New Objectives) to an Orbitrap Fusion instrument (Thermo Fisher Scientific). nLC was operated in a single analytical column set up using PicoFrit Emmiters (New Objectives, 75μm inner diameter) packed in-house with Reprosil-Pure-AQ C18 phase (Dr. Maisch, 1.9-μm particle size), with the flow rate 200 nl/min. Sample dissolved in 0.1% formic acid was injected onto the column and eluted in a gradient from 2 to 25% acetonitrile in 65 min and from 25% to 80% acetonitrile in 10 min, followed by isocratic elution at 80% acetonitrile for 15 min (total elution time 90 min). The nanoSpray ion source was operated at 2.1 kV spray voltage and 300 °C heated capillary temperature. A precursor MS1 scan (m/z 350–1700) of intact peptides was acquired in the Orbitrap at a nominal resolution setting of 120,000. For data analysis raw spectra were deconvoluted to zero charge by Bio-Pharma Finder Software (Thermo Fisher Scientific) as previously described with minor modifications[76]. Briefly, sliding windows method was used for chromatography and source spectra with target average spectrum width of 0.1 min in the range 74-76 min. ReSpect deconvolution algorithm was used for signal deconvolution. Glycoproteoforms were annotated manually from zero-charge deconvoluted intact MS data using average masses of hexose, deoxyhexose, HexNAc, and the known predicted mass of the native consomatin sequence.

## O-Glycan profiling

Subfraction # 32-20, containing glycopeptides, was dried and reconstituted in 10 μL water. O-glycan release, labeling and C18 nanoLC-MS/MS analysis was performed as described previously with minor adjustments[77]. Briefly, an in-solution nonreductive O-glycan release was performed, using 15 μL 34% hydroxylamine and 29% 1,8-diazabicyclo(5.4.0)undec-7-ene (DBU) in water[78] (75 min at 37 °C). Released glycans were purified by 2 mg of magnetic hydrazide beads (MagSi-S Hydrazide beads 1 μm, magtivio B.V., Nuth), labeled using 2-aminobenzamide, and purified by cotton hydrophilic interaction chromatography (HILIC) using solid phase extraction (SPE)[79] and further purified using porous graphitic carbon (PGC) SPE[80]. Prior to analysis the glycans were dried and reconstituted in 10 μL water. One μL per sample was separated by nanoflow liquid chromatograph using a single

analytical column setup packed with Reprosil-Pure-AQ C18 phase (Dr. Maisch, 1.9 μm in particle size, 30 cm in column length) in an EASY-nLC 1200 UHPLC (Thermo Fisher Scientific) using a PicoFrit Emitter (New Objectives, 75 μm in inner diameter). Mobile phase A was 0.1% formic acid (FA) in water, mobile phase B 0.1% FA in 80% ACN. A gradient from 2% to 38% mobile phase B in 20 min was used for elution of glycans. The emitter was interfaced to an Orbitrap Fusion Lumos MS (Thermo Fisher Scientific) via a nanoSpray Flex ion source. A precursor MS scan (m/z 275-1700, positive polarity) was acquired in the Orbitrap at a nominal resolution of 120,000, followed by Orbitrap higher-energy C-trap dissociation (HCD)-MS/MS at a nominal resolution of 30,000 of the 10 most abundant precursors in the MS spectrum (charge states 1 to 4). A minimum MS signal threshold of 50,000 was used to trigger data-dependent fragmentation events. HCD was performed with an energy of 27% ± 5%, applying a 10 s dynamic exclusion window. Data analysis was performed using Preoteome Discoverer 2.5.0.400 (Thermo Fisher Scientific Inc.) for peak picking and Skyline 22.2.0.351 (ProteoWizard) for quality control and data integration.

## Edman sequencing

N-terminal peptide sequencing of subfraction # 32-20 was conducted at the Protein Facility at Iowa State University with prior deblocking of the N-terminus. An estimated 2 nmol of purified peptide was used for sequencing. The sample was loaded onto a Shimadzu PVDF membrane and dried using the reaction vessel. The membrane was washed with methanol and DI water, then loaded onto a Shimadzu PPSQ-53A instrument for sequence analysis.

## Reporting summary

Further information on research design is available in the Nature Portfolio Reporting Summary linked to this article.

## Data availability

The sequencing data generated in this study have been deposited in the NCBI Short Sequence Archive (SRA) under the following accession number: SRR27535905 Section B4, SRR27535906 Section B3, SRR27535907 Section B2, SRR27535908 Section B1 (BioProject ID: PRJNA1064068). The mass spectrometry proteomics data have been deposited to the ProteomeXchange Consortium via the PRIDE [1] partner repository with the dataset identifier PXD053107 and 10.6019/PXD053107 (http://www.ebi.ac.uk/pride/archive/projects/PXD052869). All data supporting the findings described in this manuscript are available in the article and in the Supplementary Information. All data will be provided by the corresponding author upon request. Source data are provided with this paper.

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

## Acknowledgements

The authors thank Noel Saguil for assistance with field collections, Dylan Taylor for the images of *C. geographus* capturing fish, Paula Flórez Salcedo for the illustration of *Conus geographus*, Nicholas Schumann for the illustration of the chemical structure of pG1, the Iowa State University Protein facility for Edman sequencing. Figure 3A–C illustrations were created with BioRender.com. Plasmids encoding Venus(155–239)-Gβ1 (human); Venus(1–155)-Gγ2 (human); masGRK3ct-Nluc-HA (human); Gα$_{i1}$ (rat); Gα$_{i2}$ (rat); Gα$_{i3}$ (rat); Gα$_{oA}$ (human) and Gα$_{oB}$ (human) were kindly provided by Dr. Kirill Martemyanov, Scripps Research, FL, USA. Mass spectrometry analysis were performed by the Proteomics Research Infrastructure (PRI) at the University of Copenhagen (UCPH), supported by the Novo Nordisk Foundation (NNF) (grant agreement number NNF19SA0059305). This work was supported by a Villum Young Investigator Grant (19063 to H.S.-H.), a Starting Grant from the European Commission (ERC-Stg 949830 to H.S.-H.), a National Institutes of Health Grant (GM144719 to B.M.O. and H.S-H.), and the Carlsberg Foundation (CF20-0248 to H.B.-O.). T.L.K. is supported by a fellowship from Independent Research Fund Denmark (DFF 3102-00006B). A.H. is supported by a fellowship from Svenska Sällskapet for Medicinsk Forskning (SSMF). J.G.K. is supported by a Novo Nordisk Fonden Excellence Emerging Investigator Grant- Endocrinology & Metabolism (no. 0054300) and an Independent Research Fund Denmark Sapere Aude Fellowship (no. 0169-00067B). KTS is supported by a Novo Nordisk Fond Hallas Møller Ascending Investigator grant (no. 0073793) and a Sapere Aude Research Leader Grant from the Independent Research Fund Denmark (2066-00043B).

## Author contributions

H.Y.Y., I.B.R.L., D.B.A., T.L.K., W.E.B.-Y., J.G.K., H.B.O., K.T.S., J.J.H., H.S.-H. designed the research; H.Y.Y., I.B.R.L., D.B.A., T.L.K., A.H., S.E., S.Y.V., K.B.P., N.H., H.S.-H. performed the research; H.Y.Y., I.B.R.L., D.B.A., T.L.K., S.Y.V., K.B.P., N.H., A.L.H.E., J.G.K., K.T.S., H.S.-H. analyzed the

data; H.Y.Y., I.B.R.L., D.B.A., T.L.K., B.M.O., H.S.-H. wrote the manuscript. All authors have read and approved the manuscript.

## Competing interests

The authors declare no competing interests.
