## [Peer Review File · Nature Communications]

Fish-hunting cone snail disrupts prey's glucose homeostasis with weaponized mimetics of somatostatin and insulinREVIEWER COMMENTS

Reviewer #1 (Remarks to the Author):

Comments to Authors

In this MSS entitled 'Disruption of Glucose Homeostasis in Prey: Combinatorial Use of Weaponized Mimetics of Somatostatin and Insulin by a Fish-Hunting Cone Snail' Ho Yan Yeung, demonstrate that the toxin's N-terminal tail aligns with a glycosylated somatostatin peptide previously identified from fish pancreas and plays an important role in activating the fish SSTR2.

Authors show that, in addition to insulins, the deadly fish hunter, *Conus geographus*, uses a selective agonist of the somatostatin receptor 2 (SSTR2) that potently blocks the release of the insulin-counteracting hormone glucagon, thereby exacerbating insulin-induced hypoglycemia in prey.

Collectively, these findings provide a stunning example of chemical mimicry, highlight the combinatorial nature of venom components, and establish glucose homeostasis as an effective target for prey capture.

The concept that insulin is used by snail to induced hypoglycemic shock is not new rather a well-established concept that has been reported by authors and others before.

Authors have done enormous amount of the work to support the conclusion and manuscript is well written, data are nicely presented and organized and it was a fun to read this work. Material and Method section is well described.

Some concerns which need response from authors are as following

Figure 1 panel A has been out since 2015 by senior authors of this manuscript and source of this figure should be included.

Data presented on co-expression of con-insulins and consomatins, binding studies on Consomatin pG1 and consomatin pG1 mediated suppression of glucagon release are well organized and nicely done and results as presented were expected based on the well known function of somatostatin. The evidence for co-expression is based on RNA sequencing data. The study could be strengthened by direct evidence of co-localization within the same cells, possibly through immunohistochemistry.

A dose-dependent effect of consomatin pG1 on glucagon secretion is performed. However, the minimum and saturation doses are not clear. A wider range of dose testing could provide a more detailed dose-response curve.

Is there any significant difference in downstream pathway activation following glucagon inhibition by consomatin pG1 vs SST14? As Glucagon levels did not return to baseline levels after a 15-minute washout period, it raises questions about how consomatin pG1 and SST14 are different in terms of their mechanism of action on pancreatic function.

However, on a separate note results presented in next half part of manuscript are exciting including activation of one of the two homologous fish SSTR2 receptors, characterization of fraction # 32, endogenous somatostatin-like signaling gene that acquired a novel exon containing the glycosylated residues and Glycosylated consomatin nG1 is a motif-minimized mimetic of fish ss4.

On page 18 authors discuss some of the significance of consomatin pG1, and its selectivity to only SSTR2 out of five different receptors is important but still a limited scope as therapeutic drugs. Several studies have shown that SSTR subtypes work in a concert and this issue has not been discussed in this manuscript and some functional studies are still missing from this study.

The finding that whole venom activates both fish SSTR2 receptors while consomatin pG1 does not highlight the need to characterize the interactions between different native venom components. Modeling or docking studies could shed light on the molecular interactions responsible for the selective activation of Dr-sstr2b over Dr-sstr2a by consomatin pG1.

O-glycosylation in both venom and endogenous peptides suggests conserved evolutionary adaptations. Does this have any impact on affinity to receptor or efficacy? Comparing the activity of glycosylated versus non-glycosylated peptide forms, particularly in the context of SSTR2 receptor activation and glucagon secretion inhibition, would be beneficial.

Reviewer #2 (Remarks to the Author):

The manuscript describes a case of striking molecular mimicry by a fish-eating cone snail and provides a greater understanding of the mechanisms by which venomous animals hijack signalling systems of their prey. I thoroughly enjoyed reading the manuscript, which is very well-written with generally convincing and easy to follow arguments. The study design and method descriptions are also sound and well described. The results have clear impacts in drug discovery research but, in my opinion, first and foremost represent an utterly fascinating piece of basic biological research that I am sure will be of great interest to a wide scientific audience.

Most of my comments and suggestions are relatively minor and deal with quite specific parts of the manuscript, although I do have two more substantial suggestions that I think would improve the manuscript:

1) I realise if this is beyond what the authors consider the scope of the paper, but it would be very interesting to test the synthetic but non-glycosylated variant of nG1. If the authors have already done this, but not included any functional data because it renders the peptide virtually insoluble in biologically relevant concentrations — which is the case for some glycosylated peptides — this would still be an interesting finding and provide one of the potential functions of this modification. Conversely, if the peptide is soluble, testing its activity would provide a functional context to this PTM, provide insight into the selective forces that were involved in the emergence and fixing of the new part of the G1 consomatins (solubility would be an arguably stronger selective force than tuning activity of overall peptide), and potentially give very useful insights into native SSTR signalling.

2) Based on my personal opinion, I would recommend that the authors consider revising their introduction and discussion somewhat.

As far as I see it, a major finding is the mechanism by which *C. geographus* has mimicked fish somatostatin and I think the introduction would benefit from introducing the concept of doppelganger toxins and how these have evolved in venomous animals, particularly in cone snails but perhaps also other lineages such as ants. This would provide a better context for understanding the backdrop of the evolution of G1 and G2, as well as how the mechanism by which they have evolved is new.

Similarly, while the discussion is interesting and comprehensive, I found its current form perhaps too detailed on the various other doppelganger toxins that have been described and maybe too speculative on the benefits of having identified the *C. geographus* consomatins and other highly selective bioactive peptides earlier. For example, the paragraph on exendin seems quite extensive given its apparent aim is to suggest, and establish, glucose homeostasis as a target for venoms — an hypothesis I think is well warranted. Instead, I think the discussion would benefit from condensing the literature review on doppelganger toxins and retaining a bit more focus on the mechanisms of molecular mimicry, which in my opinion would align better with the results.

SPECIFIC COMMENTS (line numbers would have made this easier):

- Abstract should not contain references

- Second to last paragraph page 4: The authors should maybe consider rewriting this paragraph, which essentially summarises the entire paper, to a form that more concisely summarises the overall rationale behind the approach and the general findings. Reading it I am left with questions as to why the different experiments were done the way they were (e.g., why is co-expression informative?). Although these are all answered further down in the manuscript I found it a bit distracting.

- Halfway in second to last paragraph page 4: (...)the native toxin contains a heavily O-glycosylated N-terminus tail THAT plays (...)

- Last sentence page 6: Consider switching the order of octreotide and SS-14 activation values to match the sentence for improved readability.

- Last sentence page 9: The semicolon seems like it should be a colon? Should it be "to close to null"? Otherwise it sounds like there is no change, which is quite opposite to the effect.

- End of last paragraph on page 14: I apologise if I missed this, but how do the authors know whether this is an allelic variant or a third paralog? Or conversely, if only one locus was found in the genomic data, could all three be allelic variants?

Fig. S13 is difficult to read. It may be better to have this as a table in a more readable format.

- End of first paragraph page 16: I think it is a bit too soon to conclude that this is a de novo birth in the *C. Geographic* genome. It is certainly a possibility, or alternatively possibly a quite recent event, but to determine the timing of this event one needs to do a microsynteny analysis with several of the most closely related species of *Conus*.

When aligning the novel exon with the intron of the paralogs of G1, are there any signs that give an indication of the potential for novel exon inclusion? Are there, for example, regions of short tandem repeats?

The final sentence seems like it should be the first sentence of the following sub-section/paragraph: Given the selective evolution of this region and its importance in receptor activation we next investigated similarities between the native venom peptides and the endogenous fish hormone(s).

- Currently second sentence in second paragraph page 16: Sequence alignments are comparative. Remove "Comparative" from "Comparative sequence alignment"

- Fig. 7: B) these are aligned exons. The proportion of exon 2 that codes for the mature peptide seems much longer in G1 than both the other peptides. C) It would be useful to include the endogenous SSRP-like peptide from *C. geographus* in the alignment.

- First word of the discussion: This is purely a semantic comment, but optimised implies that venom traits are always at global optima when they in fact could be relatively speaking far from them. For example, they (or some or many of them) could be i) stuck at local optima, ii) constrained by e.g. peiotropic effects so that they never reach their theoretical optima, or iii) never reach constantly shifting optima (e.g. because of antagonistic co-evolution). Maybe consider using "Tuned" instead?

- End of first sentence of discussion ("to incapacitate prey and predators"): Consider including "deter predators". Predators don't have to be incapacitated, only deterred.

- Middle of second paragraph page 18 ("a novel exon "): See above comment regarding timing and mechanism of exon inclusion.

- Middle of second paragraph page 18 ("the most striking example"): I don't think this statement of novelty is necessary. The findings are fascinating irrespective of whether or not they are THE MOST striking example of mimicry reported to date (which is debatable, although this example is certainly up there).

- Beginning of second paragraph page 18 ("alternatives to man-made drugs"): I think I understand the point that the authors are trying to make, but this sentence in some ways suggests that compounds extracted from natural sources are good alternatives to synthesised compounds. Maybe consider replacing "man-made" with "human-engineered" to make it more clear that natural compounds can be excellent sources of therapeutics, as information not isolated compounds? Or reword to state that it highlights the value of naturally sourced compounds to inform engineering of drugs?

- Second paragraph page 19: Should be colon instead of semicolon in "role as weaponized hormones;"

- End of second paragraph page 20 ("existence of O-glycosylated threonine residues"): Without dismissing its potential role in receptor binding selectivity, glycosylation could also be a mechanism for improving solubility of a peptide with a very hydrophobic tail. As stated above, this could be addressed by synthesising nG1 without glycosylation.

Reviewer #3 (Remarks to the Author):

In this very interesting and original study, Yeung et al. revealed the combinatorial use by the cone snail *C. geographus* of two venom compounds, con-insulin and consomatin, that mimic the actions of insulin and somatostatin, respectively, to induce a hypoglycemic shock in prey.

In their work, the authors demonstrated the co-expression of co-insulins and consomatins in the conus venom gland. They showed that one of the consomatins, namely consomatin pG1, acts as a very highly selective SSTR2 agonist, that can inhibit glucagon secretion in mouse islets and perfused rat pancreas. They were also able to isolate the native peptide and they revealed that it exists in several forms whose the N-terminal region is heavily glycosylated. Finally they found that the toxin's N-terminal tail aligns with a glycosylated somatostatin peptide previously isolated from catfish.

The study has been particularly well conducted with great care and appropriate methods. The results have been analyzed in detail and their interpretation is conceptually sound. I also really appreciated the section discussing with honesty the limitations of the study.

My only regret is that the authors did not test the action of Consomatin in a teleost system. It is also a pity that they did not try to synthesize a glycosylated form of Consomated. However, I am not a chemist, so I don't realize how difficult it is to produce such a molecule, whose exact structure has not yet been fully elucidated.

This reservation notwithstanding, my opinion is that this work deserves to be published in Nature Communications after following minor matters are dealt with properly.

Line 80 : a word is lacking (causes ?)

Line 84. Add « In mammals ».

Line 148. Specify : « for consomatin from *C. rolani* (R01) »

Line 185 and Fig. 1C. Please explain how the structure of the mature peptides was predicted.

Line 308. The term « marine fish » is too vague. Please specify.

Line 419. Please enlarge the arrows (poorly visible)

Line 432. « ... comparison of Consomatin G1 and G2 (Fig. 7A) ». There is a problem in this sentence since the sequence of Consomatin G2 is not displayed in fig. 7A

Line 455. An important feature of SS4 that could be mentioned by the authors is that it exhibits a very variable structure among teleost species (discussed in ref. 30). This is in strong contrast with the other peptides of the SS family whose sequence has been generally strongly conserved (e.g. SS1 structure is exactly the same from lampreys to mammals).

Line 507 : suggest

REVIEWER COMMENTS

Reviewer #1 (Remarks to the Author):

In this MSS entitled 'Disruption of Glucose Homeostasis in Prey: Combinatorial Use of Weaponized Mimetics of Somatostatin and Insulin by a Fish-Hunting Cone Snail' Ho Yan Yeung, demonstrate that the toxin's N-terminal tail aligns with a glycosylated somatostatin peptide previously identified from fish pancreas and plays an important role in activating the fish SSTR2.

Authors show that, in addition to insulins, the deadly fish hunter, *Conus geographus*, uses a selective agonist of the somatostatin receptor 2 (SSTR2) that potently blocks the release of the insulin-counteracting hormone glucagon, thereby exacerbating insulin-induced hypoglycemia in prey.

Collectively, these findings provide a stunning example of chemical mimicry, highlight the combinatorial nature of venom components, and establish glucose homeostasis as an effective target for prey capture.

The concept that insulin is used by snail to induced hypoglycemic shock is not new rather a well-established concept that has been reported by authors and others before.

Authors have done enormous amount of the work to support the conclusion and manuscript is well written, data are nicely presented and organized and it was a fun to read this work. Material and Method section is well described.

Some concerns which need response from authors are as following

Comment 1: Figure 1 panel A has been out since 2015 by senior authors of this manuscript and source of this figure should be included.

Response: We understand that the images used in Figure 1, panel A are similar to another image we used in our 2015 study. The pictures used in this current panel have not yet been published in a scientific journal but have been used in several media reports. We have now provided information on the photographer who took the original pictures in the revised figure caption ("Photographs taken by Dylan Taylor", line 173).

Comment 2: Data presented on co-expression of con-insulins and consomatins, binding studies on Consomatin pG1 and consomatin pG1 mediated suppression of glucagon release are well organized and nicely done and results as presented were expected based on the well known function of somatostatin.

The evidence for co-expression is based on RNA sequencing data. The study could be strengthened by direct evidence of co-localization within the same cells, possibly through immunohistochemistry.

Response: We agree with the reviewer that a more fine-grained analysis of the co-expression patterns would have been informative. To address this, we attempted performing single cell transcriptome sequencing which could have provided information on the potential co-expression of toxins in the same cells. However, we currently do not have enough biological material (i.e., fresh or appropriately preserved venom glands of *C. geographus*) to optimize this method or establish alternative methods such as in situ hybridization or immunohistochemistry in our lab. We have now included a sentence on the limitation of the current data in addressing c-expression and potential coordinated release of the two toxins from the same cells (lines 168-169).

Comment 3a: A dose-dependent effect of consomatin pG1 on glucagon secretion is performed. However, the minimum and saturation doses are not clear. A wider range of dose testing could provide a more detailed dose-response curve.

Response: We agree with the reviewer that a wider range of dose testing could have provided a more detailed picture of the dose-response curve. However, we would like to note that rather than showing a detailed dose-response we aimed at providing evidence to show that the toxin can inhibit pancreatic glucagon secretion in a dose-dependent manner. Given the biological variations observed in this model, obtaining a detailed dose-response curve beyond the three doses that were already tested would have required many more animals. We felt that this was not justifiable given that the current data in both islets and pancreas perfusion studies (Fig. 3B and 3C) already supports our hypothesis.

Comment 3b: Is there any significant difference in downstream pathway activation following glucagon inhibition by consomatin pG1 vs SST14? As glucagon levels did not return to baseline levels after a 15-minute washout period, it raises questions about how consomatin pG1 and SST14 are different in terms of their mechanism of action on pancreatic function.

Response: While consomatin pG1 and SST14 may have different mechanistic effects on SSTR₂-associated pancreatic function we note that application of 10 nM of SST-14 shows a comparable effect on glucagon secretion as consomatin pG1 (glucagon secretion does not return to baseline levels after a 15 min wash-out period, see Xu et al., *Acta Physiol (Oxf)*, 2020 Jul;229(3):e13464, Fig. 2A.). We have now added this additional information and reference to the prior study in the revised manuscript (lines 283-286). However, we point out that a direct comparison between pG1 and SST-14 using the perfused pancreas model is difficult due to the fact that SST-14 also binds to the SSTR₅ and is likely less stable than the venom peptide.

Comment 4: However, on a separate note results presented in next half part of manuscript are exciting including activation of one of the two homologous fish SSTR2 receptors, characterization of fraction # 32, endogenous somatostatin-like signaling gene that acquired a novel exon containing the glycosylated residues and Glycosylated consomatin nG1 is a motif-minimized mimetic of fish ss4.

Response: We agree and thank the reviewer for this comment.

Comment 5: On page 18 authors discuss some of the significance of consomatin pG1, and its selectivity to only SSTR2 out of five different receptors is important but still a limited scope as therapeutic drugs. Several studies have shown that SSTR subtypes work in a concert and this issue has not been discussed in this manuscript and some functional studies are still missing from this study.

Response: We thank the reviewer for this comment and have revised this paragraph to clarify the usefulness of multireceptor-targeted SSTR agonists (lines 549-552 and line 558). Indeed, a selective SSTR₂ agonist, such as the one described here, does not necessarily represent the best ligand for SSTR₂-expressing neoplasms and cancer.

Comment 6: The finding that whole venom activates both fish SSTR2 receptors while consomatin pG1 does not highlight the need to characterize the interactions between different native venom components. Modeling or docking studies could shed light on the molecular interactions responsible for the selective activation of Dr-sstr2b over Dr-sstr2a by consomatin pG1.

Response: We agree that it would have been interesting to investigate the molecular interactions responsible for the selective activation of Dr-sstr2b over Dr-sstr2a by consomatin pG1 by molecular docking and structural studies. We attempted to elucidate potential ligand-receptor interactions using molecular docking of pG1 with the Dr-sstr2a and Dr-sstr2b built on a

hSSTR2 homology model and an alpha-fold predicted model with N-terminus truncations of the fish receptors. Unfortunately, the results did not reveal any obvious differences between the toxin's potential interaction with the two fish receptors, particularly with the TM bundles and the ECL2 which are known to be critical for ligand interaction and receptor activation. However, the ECL2 regions are highly dynamic and poorly resolved by alpha fold ($70 > \text{pLDDT} > 50$) possibly explaining why docking studies could not reveal differences in the ability of pG1 to activate the two fish receptors.

To further address this, we have begun to perform mutagenesis studies on the Dr-sstr2a. Our preliminary results show that inserting the ECL2 region of the hSSTR₂ into the Dr-sstr2a results in activation of this hybrid receptor by pG1. While these are intriguing preliminary results, we strongly feel that additional work is needed to fully elucidate the structural and functional differences between the human and the two fish receptors. These studies are beyond the scope of our current manuscript and will be subject of a follow-up study.

Comment 7: O-glycosylation in both venom and endogenous peptides suggests conserved evolutionary adaptations. Does this have any impact on affinity to receptor or efficacy?

Comparing the activity of glycosylated versus non-glycosylated peptide forms, particularly in the context of SSTR2 receptor activation and glucagon secretion inhibition, would be beneficial.

Response: We agree that the presence of the glycosylation raises questions about the role of this modification for activation of the fish SSTR₂. To address this, we tested an analog of nG1 that lacks the glycosylation (des-glyco nG1) in comparison with the venom fraction, # 32-20, containing the glycosylated proteoforms. These additional studies are shown in Figs. S15 and S16 and the result section of the revised manuscript (lines 434-456). While the non-glycosylated version of nG1 acts as a partial agonist at the Dr-sstr2a and partially rescues the activation of the Dr-sstr2a when compared to pG1, this analog cannot recapitulate the activation of this receptor mediated by #32-20. This strongly suggests that the glycosylation is needed for full functional activity at the Dr-sstr2a which may explain the presence of this modification in both, the endogenous cat fish peptide and the venom peptide.

Based on the following findings we decided not to further test the non-glycosylated analog using pancreatic perfusion assays or islets: des-glyco nG1 is only slightly less potent at activating the hSSTR2 when compared to SS-14 (Fig. S16) and is several orders of magnitude more potent at this receptor than any other human SSTR subtype. Thus, given that des-glyco nG1 activates the hSSTR2 comparable to SS-14, we do not expect major differences on glucagon secretion between this peptide and pG1 in the rodent pancreas assay but would potentially expect differences in a teleost model (not tested here).

Reviewer #2 (Remarks to the Author):

The manuscript describes a case of striking molecular mimicry by a fish-eating cone snail and provides a greater understanding of the mechanisms by which venomous animals hijack signalling systems of their prey. I thoroughly enjoyed reading the manuscript, which is very well-written with generally convincing and easy to follow arguments. The study design and method descriptions are also sound and well described. The results have clear impacts in drug discovery research but, in my opinion, first and foremost represent an utterly fascinating piece of basic biological research that I am sure will be of great interest to a wide scientific audience.

Most of my comments and suggestions are relatively minor and deal with quite specific parts of the manuscript, although I do have two more substantial suggestions that I think would improve the manuscript:

Comment 1:

1) I realise if this is beyond what the authors consider the scope of the paper, but it would be very interesting to test the synthetic but non-glycosylated variant of nG1. If the authors have already done this, but not included any functional data because it renders the peptide virtually insoluble in biologically relevant concentrations — which is the case for some glycosylated peptides — this would still be an interesting finding and provide one of the potential functions of this modification. Conversely, if the peptide is soluble, testing its activity would provide a functional context to this PTM, provide insight into the selective forces that were involved in the emergence and fixing of the new part of the G1 consomatatin (solubility would be an arguably stronger selective force than tuning activity of overall peptide), and potentially give very useful insights into native SSTR signalling.

Response: We agree with the reviewer that comparative analysis using the non-glycosylated peptide forms could provide additional information on the potential role of the glycans in both the venom peptide and endogenous fish ss4 peptides. To address this, we tested the non-glycosylated form of nG1 glycosylation (des-glyco nG1) at the fish and human receptors (see Figs. S15-16 and updated method and result section, lines 434-456). While the non-glycosylated version of nG1 partially rescues the activation of the Dr-sstr2a observed by the venom fraction, this analog cannot recapitulate full receptor activation. This strongly suggests that the glycosylation is needed for full functional activity at the Dr-sstr2a which may explain the presence of this modification in both, the endogenous cat fish peptide and the venom peptide.

Furthermore, while glycosylations can also aid in increasing the solubility of peptides in aqueous solutions, as previously demonstrated for the peptide Mg7a from the venom of the giant red bull ant (Robinson et al., 2021), we did not observe any obvious differences in the solubility of des-glyco nG1, pG1, or the venom fraction. However, given the complex nature of the venom peptide forms we were unable to generate synthetic versions of the glycosylated forms for direct, side-by-side comparison. Thus, as discussed in the revised text (lines 453-456) whether the observed functional differences between des-glyco nG1 and the purified venom fraction at the Dr-sst2a are purely based on the presence of the glycans remains to be determined.

Comment 2:

2) Based on my personal opinion, I would recommend that the authors consider revising their introduction and discussion somewhat.

As far as I see it, a major finding is the mechanism by which *C. geographus* has mimicked fish somatostatin and I think the introduction would benefit from introducing the concept of doppelganger toxins and how these have evolved in venomous animals, particularly in cone snails but perhaps also other lineages such as ants. This would provide a better context for

understanding the backdrop of the evolution of G1 and G2, as well as how the mechanism by which they have evolved is new.

Similarly, while the discussion is interesting and comprehensive, I found its current form perhaps too detailed on the various other doppelganger toxins that have been described and maybe too speculative on the benefits of having identified the *C. geographus* consomatins and other highly selective bioactive peptides earlier. For example, the paragraph on exendin seems quite extensive given its apparent aim is to suggest, and establish, glucose homeostasis as a target for venoms — an hypothesis I think is well warranted. Instead, I think the discussion would benefit from condensing the literature review on doppelganger toxins and retaining a bit more focus on the mechanisms of molecular mimicry, which in my opinion would align better with the results.

Response: We have now introduced the concept of doppelganger peptides in the introduction (lines 72-86) and revised our discussion to provide additional examples of doppelganger peptides from other venomous animals (lines 592-594). We decided not to include a more thorough review of molecular mimicry in this current manuscript but are considering expanding on this topic in a separate review paper. We have left the paragraph on the gila monster peptide in the discussion section as it represents one of the very few functionally-validated examples of a toxin that modulates glucose homeostasis. However, we have now shortened this section in the revised version (lines 575-586).

SPECIFIC COMMENTS (line numbers would have made this easier):

Note by the authors: We apologize for the lack of line numbers. The version we submitted had line numbers which must have been removed during the submission process.

Comment 3: Abstract should not contain references

Response: This has been fixed.

Comment 4: Second to last paragraph page 4: The authors should maybe consider rewriting this paragraph, which essentially summarises the entire paper, to a form that more concisely summarises the overall rationale behind the approach and the general findings. Reading it I am left with questions as to why the different experiments were done the way they were (e.g., why is co-expression informative?). Although these are all answered further down in the manuscript I found it a bit distracting.

Response: We have rewritten this paragraph to hopefully more concisely summarize the rationale and findings of our study (lines 125-139).

Comment 5: Halfway in second to last paragraph page 4: (...)the native toxin contains a heavily O-glycosylated N-terminus tail THAT plays (...)

Response: This paragraph has been rewritten (see comment 4 above).

Comment 6: Last sentence page 6: Consider switching the order of octreotide and SS-14 activation values to match the sentence for improved readability.

Response: The order has been switched.

Comment 7: Last sentence page 9: The semicolon seems like it should be a colon? Should it be "to close to null"? Otherwise it sounds like there is no change, which is quite opposite to the effect.

Response: This has been fixed.

Comment 8: End of last paragraph on page 14: I apologise if I missed this, but how do the

authors know whether this is an allelic variant or a third paralog? Or conversely, if only one locus was found in the genomic data, could all three be allelic variants?

Response: When assessing cone snail transcriptomes through mapping reads back onto assembled transcripts, we almost always observe two highly similar sequences that only differ by 1-2 nucleotides across the entire sequence and are usually present in a 1:1 ratio. Based on this we think that the variant observed by mass spec (Fig. S12) and also by mapping reads back onto transcripts (Fig. S1) is an allelic variant rather than another paralog. We have revised the text to now read "... that only differs from Consomatin nG2 by a single amino acid, likely representing an allelic variant of G2 (nG2ii) (Fig. 6B, Supp. Fig. S1A, S12-S14)." Lines 409-410

Comment 9: Fig. S13 is difficult to read. It may be better to have this as a table in a more readable format.

Response: This figure has been changed to make it more readable.

Comment 10: End of first paragraph page 16: I think it is a bit too soon to conclude that this is a de novo birth in the *C. Geographus* genome. It is certainly a possibility, or alternatively possibly a quite recent event, but to determine the timing of this event one needs to do a microsynteny analysis with several of the most closely related species of *Conus*.

Response: It is correct, that the exact timing of the de novo birth of the additional exon found in *C. geographus* G1 gene is impossible to establish without genomic information from closely related species. Currently, no such data is available. However, RNAseq from two closely related species, *C. tulipa* and *C. obscurus*, does not reveal any transcripts with the characteristic N-terminal extension of Consomatin G1 suggesting that the de novo exon birth may be specific to *C. geographus*. However, it is possible that the identified transcripts are paralogs or alternatively spliced isoforms. Genomic data from *C. ventricosus*, *C. betulinus*, and *C. canariensis* only identifies three exons and 12 other phylogenetically diverse cone snail species similarly only encode 'canonical' consomatins of three exons (Phuong 2018 <https://www.ncbi.nlm.nih.gov/pmc/articles/PMC5913681>).

To address the reviewer's concern, we have now deleted "in the *C. geographus* genome" from the original sentence in line 476 ("...originated from the de novo birth of a novel exon 3 in the *C. geographus* genome"). Furthermore, we have rephrased the original sentence "...that is encoded by a novel exon in the *C. geographus* genome" to "that is encoded by an additional exon in the *C. geographus* genome" (lines 471 and 509).

Comment 11: When aligning the novel exon with the intron of the paralogs of G1, are there any signs that give an indication of the potential for novel exon inclusion? Are there, for example, regions of short tandem repeats?

Response:

We have not been able to detect any unusual number of microsatellite repeats, or transposable elements in the intronic regions. Alignments of the intron/novel exon region across *C. ventricosus*, *C. betulinus*, and *C. geographus* show many highly conserved regions, including the region encoding the novel exon. Based on this alignment it is most likely that an intronic region was exonized in *C. geographus* (or in a close ancestor of *C. geographus*) and that the most significant factor in the exonization was a ~350 bp deletion in *C. geographus* (or in a close ancestor of *C. geographus*) bringing the donor splice site closer to this region. However, the short length of the *C. geographus* scaffold makes it difficult to interpret the alignment downstream of the novel exon into the *C. geographus* exon 3.

Comment 12: The final sentence seems like it should be the first sentence of the following subsection/paragraph: Given the selective evolution of this region and its importance in receptor activation we next investigated similarities between the native venom peptides and the

endogenous fish hormone(s).

Response: This sentence has been moved to the start of the next paragraph.

Comment 13: Currently second sentence in second paragraph page 16: Sequence alignments are comparative. Remove "Comparative" from "Comparative sequence alignment"

Response: The word "Comparative" has been removed.

Comment 14: Fig. 7: B) these are aligned exons. The proportion of exon 2 that codes for the mature peptide seems much longer in G1 than both the other peptides. C) It would be useful to include the endogenous SSRP-like peptide from *C. geographus* in the alignment.

Response: The proportion of exon 2 encoding the mature G1 is right. We are limited in the number of genomes we can assess. The predicted toxins vary in length, and several have longer N-termini extending into exon 2.

We have added the endogenous SSRP-like peptide from *C. geographus* in the revised figure.

Comment 15: First word of the discussion: This is purely a semantic comment, but optimised implies that venom traits are always at global optima when they in fact could be be relatively speaking far from them. For example, they (or some or many of them) could be i) stuck at local optima, ii) constrained by e.g. peiotropic effects so that they never reach they theoretical optima, or iii) never reach constantly shifting optima (e.g. because of antagonistic co-evolution). Maybe consider using "Tuned" instead?

Response: We have changed the word to "Tuned" as it is indeed a much better term to be used here.

Comment 16: End of first sentence of discussion ("to incapacitate prey and predators"): Consider including "deter predators". Predators don't have to be incapacitated, only deterred.

Response: We have added the word "deter" to this sentence.

Comment 17: Middle of second paragraph page 18 ("a novel exon "): See above comment regarding timing and mechanism of exon inclusion.

Response:

The term "novel" has been changed to "additional" in this sentence (see answer to comment #11).

Comment 18: Middle of second paragraph page 18 ("the most striking example"): I don't think this statement of novelty is necessary. The findings are fascinating irrespective of whether or not they are THE MOST striking example of mimicry reported to date (which is debatable, although this example is certainly up there).

Response: We have changed this sentence to "one of the most...".

Comment 19: Beginning of second paragraph page 18 ("alternatives to man-made drugs"): I think I understand the point that the authors are trying to make, but this sentence in some ways suggests that compounds extracted from natural sources are good alternatives to synthesised compounds. Maybe consider replacing "man-made" with "human-engineered" to make it more clear that natural compounds can be excellent sources of therapeutics, as information not isolated compounds? Or reword to state that it highlights the value of naturally sourced compounds to inform engineering of drugs?

Response: We had not thought about the implications of "man-made" pointed out by the reviewer and have now changed this wording to "human-engineered".

Comment 20: Second paragraph page 19: Should be colon instead of semicolon in "role as

weaponized hormones;"

Response: This has been fixed.

Comment 21: End of second paragraph page 20 ("existence of O-glycosylated threonine residues"): Without dismissing its potential role in receptor binding selectivity, glycosylation could also be a mechanism for improving solubility of a peptide with a very hydrophobic tail. As stated above, this could be addressed by synthesising nG1 without glycosylation.

Response: Please see our response to comment #1.

Reviewer #3 (Remarks to the Author):

In this very interesting and original study, Yeung et al. revealed the combinatorial use by the cone snail *C. geographus* of two venom compounds, con-insulin and consomatins, that mimic the actions of insulin and somatostatin, respectively, to induce a hypoglycemic shock in prey. In their work, the authors demonstrated the co-expression of co-insulins and consomatins in the conus venom gland. They showed that one of the consomatins, namely consomatins pG1, acts as a very highly selective SSTR2 agonist, that can inhibit glucagon secretion in mouse islets and perfused rat pancreas. They were also able to isolate the native peptide and they revealed that it exists in several forms whose the N-terminal region is heavily glycosylated. Finally, they found that the toxin's N-terminal tail aligns with a glycosylated somatostatin peptide previously isolated from catfish.

The study has been particularly well conducted with great care and appropriate methods. The results have been analyzed in detail and their interpretation is conceptually sound. I also really appreciated the section discussing with honesty the limitations of the study.

Comment 1: My only regret is that the authors did not test the action of Consomatins in a teleost system. It is also a pity that they did not try to synthesize a glycosylated form of Consomatins. However, I am not a chemist, so I don't realize how difficult it is to produce such a molecule, whose exact structure has not yet been fully elucidated.

Response: We thank the reviewer for the positive assessment of our study. We agree that testing the native venom peptide in a teleost system would have added significant value to this study. We indeed sought to establish such model, based on a previous zebrafish model of diabetes that we have developed in our lab. However, we found that it is very challenging to continuously monitor glucose levels in zebrafish and that fish have to be sacrificed to obtain single dose and single time point values. Using this set-up would have required hundreds of adult zebrafish. Alternatively, we investigated the use of larger fish species for which continuous glucose monitoring could have been more feasible. We currently do not have sufficient venom to carry out these studies. Having a synthetic version of the venom peptide would have solved this problem and could have also allowed us to better investigate the role of the complex glycan(s) for receptor activation. However, currently synthesizing such complex structures is not feasible. To somewhat address the reviewer's critique, we now show that the venom fraction is capable of inhibiting glucagon secretion in perfused rat pancreas (Supp. Fig. S10, lines 359-360 in revised text). While this is not a teleost model, the existence of a homologous system in fish combined with our observation that the venom peptide closely mimics a somatostatin peptide previously isolated from the pancreas of the cat fish makes it likely that our experiments in rodents are translatable to a teleost system.

This reservation notwithstanding, my opinion is that this work deserves to be published in Nature Communications after following minor matters are dealt with properly.

Comment 2: Line 80 : a word is lacking (causes ?)

Response: We have added the word "cause" in "that is characterized by the release of toxins into the water that induce hypoactivity and cause sedation"

Comment 3: Line 84. Add « In mammals ».

Response: We have added "in prey" as *Conus geographus* preys on fish.

Comment 4: Line 148. Specify : « for consomatins from *C. rostratus* (R01) »

Response: We have changed this to “As previously demonstrated for Consomatin Ro1 isolated from the venom of *Conus rolandi*”

Comment 5: Line 185 and Fig. 1C. Please explain how the structure of the mature peptides was predicted.

Response: We have added clarifying text to the revised manuscript: “The mature peptide was predicted based on previous findings of proteolytic cleavage and modification sites of the venom peptide Consomatin Ro1¹⁷”.

Comment 6: Line 308. The term « marine fish » is too vague. Please specify.

Response: We have replaced “marine fish” with “euteleosts”

Comment 7: Line 419. Please enlarge the arrows (poorly visible)

Response: We have enlarged the arrows in the revised Fig. 6B.

Comment 8: Line 432. « ... comparison of Consomatin G1 and G2 (Fig. 7A)». There is a problem in this sentence since the sequence of Consomatin G2 is not displayed in fig. 7A

Response: We have removed G2 from this sentence.

Comment 9: Line 455. An important feature of SS4 that could be mentioned by the authors is that it exhibits a very variable structure among teleost species (discussed in ref. 30). This is in strong contrast with the other peptides of the SS family whose sequence has been generally strongly conserved (e.g. SS1 structure is exactly the same from lampreys to mammals).

Response: We have added this additional interesting information to the revised text (lines 483-485).

Comment 10: Line 507 : suggest

Response: We have changed “suggests” to “suggest”

REVIEWERS' COMMENTS

Reviewer #1 (Remarks to the Author):

No Comments are required for authors

Reviewer #2 (Remarks to the Author):

The authors have addressed all my suggestions and comments and I look forward to seeing this very interesting manuscript in its final published form.

Reviewer #3 (Remarks to the Author):

I am satisfied with the answers given by the authors. In my opinion, their manuscript now deserves to be published as it is.